# CroMA: Cross-Modality domain Adaptation for Monocular BEV Perception

## Abstract

Incorporating multiple sensor modalities and closing the domain gap between training and deployment are two challenging yet critical topics for self-driving. Existing adaption work only focuses on the visual-level domain gap, overlooking the sensor-type gap which exists in reality. A model trained with a collection of sensor modalities may need to run on another setting with less types of sensors available. In this work, we propose a Cross-Modality Adaptation (CroMA) framework to facilitate the learning of a more robust monocular bird's-eye-view (BEV) perception model, which transfers the point clouds knowledge from a LiDAR sensor during the training phase to the camera-only testing scenario. The absence of LiDAR during testing negates the usage of it as model input. Hence, our key idea lies in the design of (i) a LiDAR-teacher and Camera-student knowledge distillation model, and (ii) a multi-level adversarial learning mechanism, which adapts and aligns the features learned from different sensors and domains. This work results in the first open analysis of cross-domain perception and cross-sensor adaptation for monocular 3D tasks in the wild. We benchmark our approach on large-scale datasets under a wide range of domain shifts and show state-of-the-art results against various baselines.

## 1 Introduction

In recent years, multi-modality 3D perception has shown outstanding performance and robustness over its single-modality counterpart, achieving leading results for various 3D perception tasks (Vora et al., 2020; Qi et al., 2020; Jaritz et al., 2020; Park et al., 2021; Weng et al., 2020) on large-scale multi-sensor 3D datasets (Caesar et al., 2020; Kesten et al., 2019; Sun et al., 2020). Despite the superiority in information coverage, the introduction of more sensor modalities also poses additional challenges to the perception system. On one hand, generalizing the model between datasets becomes hard because each sensor has its unique domain gap, such as field-of-view (FoV) for cameras, density for LiDAR, etc. On the other hand, the operation of the model is conditioned on the presence and function of more sensors, making it hard to work on autonomous agents with less sensor types or under sensor failure scenarios.

More specifically, **transferring knowledge** among different data domains is still an open problem for autonomous agents in the wild. In the self-driving scenario, training the perception models offline in a source domain with annotations while deploying the model in another target domain without annotations is very common in practice. As a result, the model will have to consider the domain gap between source and target environments or datasets, which usually involves different running locations, different sensor specifications, different illumination and weather conditions, etc.

Meanwhile, the domain shift lies not only in the **visual perspective**, but also in the **sensor-modality perspective**. Previous methods assume a less *realistic* setting where all sensors are available during training, validation, and deployment time, which is not always true in reality. Due to the cost and efficiency trade-off, or sensor missing and failure, in many scenarios we can have fewer sensors available in the target domain during testing than what we have in the source domain during training. A typical scenario is having camera and LiDAR sensors in the large-scale training phase while only having cameras for testing, as shown in Figure 1. It is not clear how to facilitate the camera-only 3D inference with the help of a LiDAR sensors only in the source domain during training.

The challenges above raise an important question and task: *Can we achieve robust 3D perception under both the visual domain gap and sensor modality shift?*

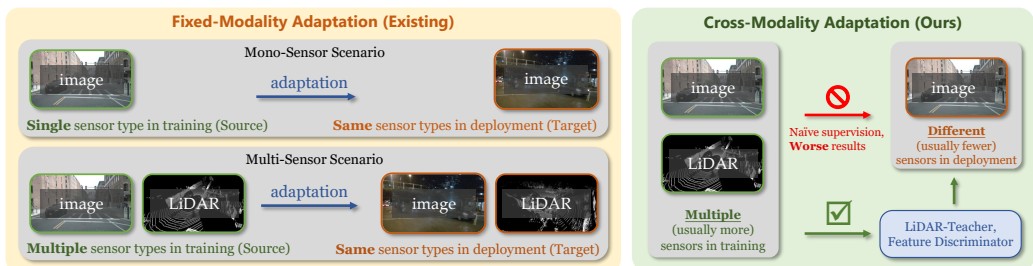

Figure 1: **Left & Middle**: Existing models assume fixed sensor modalities during training and testing phases. **Right**: We introduce a more realistic setting which considers possible cross-modality domain shift between model training and deployment. **Surprisingly**, using source-only LiDAR as depth supervision leads to worse performance (**6.3** in IoU) than the image-only model (**6.7**). Thus, we propose CroMA to reduce the domain discrepancy with knowledge distillation and feature alignment, achieving state-of-the-art performance (**17.0**).

This task is challenging for several main reasons. (1) ***Ill-posed***. The monocular 3D perception in the target domain is inherently ill-posed, due to the lack of direct 3D measurement from the camera-only sensor input. (2) ***Partial availability of LiDAR data***. Most prior work on multi-modal 3D perception (Vora et al., 2020; Man et al., 2021) and 3D domain adaptation (Zhang et al., 2021; Yang et al., 2021) makes LiDAR point clouds the input signals to their models for providing accurate range measurement. However, these methods fail when the LiDAR sensor is unavailable during the evaluation, as no LiDAR data can be used as input into the model. Hence, our new setting requires a model to leverage information from the LiDAR point clouds without making it an input to the model. (3) ***Naive LiDAR supervision leads to worse performance***. It is generally believed in the community that introducing additional sensors is bound to increase the overall performance. Surprisingly, our experiments showed 0.3 IoU *decrease* when we naively introducing LiDAR to supervise the depth estimation. This is because the source and target domain gap becomes larger with the additional sensor-type shift. As we will discuss in Sec. 3.2, our new problem setting requires novel methodology in using LiDAR without increasing the domain discrepancy.

To tackle the above challenges, we propose CroMA, a cross-modality domain adaptation framework for bird's-eye-view (BEV) perception. Our model addresses the monocular 3D perception task between different domains, and utilizes additional modalities in the source domain to facilitate the evaluation performance. Motivated by the fact that image and BEV frames are bridged with 3D representation, we first design an efficient backbone to perform 3D depth estimation followed by a BEV projection. Then, to learn from point clouds without explicitly taking them as model inputs, we propose an implicit learning strategy, which distills 3D knowledge from a LiDAR-Teacher to help the Camera-Student learn better 3D representation. Finally, in order to address the visual domain shift, we introduce adversarial learning on the student to align the features learned from source and target domains. Supervision from the teacher and feature discriminators are designed at multiple layers to ensure an effective knowledge transfer. By considering the domain gap and effectively leveraging LiDAR point clouds in the source domain, our proposed method is able to work reliably in more complicated, uncommon, and even unseen environments. Our model achieves state-of-the-art performance in four very different domain shift settings. Extensive ablation studies are conducted to investigate the contribution of our proposed components, the robustness under different changes, as well as other design choices.

The main contributions of this paper are as follows. (1) We introduce modality mismatch, an overlooked but realistic problem setting in 3D domain adaptation in the wild, leading to a robust camera-only 3D model that works in complicated and dynamic scenarios with minimum sensors available. (2) We propose a novel LiDAR-Teacher and Camera-Student knowledge distillation model, which considerably outperforms state-of-the-art LiDAR supervision methods. (3) Extensive experiments in challenging domain shift settings demonstrate the capability of our methods in leveraging source domain point cloud information for accurate monocular 3D perception.

## 2  RELATED WORK

### 2.1  MULTI-MODALITY AND CROSS-MODALITY 3D PERCEPTION

Considerable research has examined leveraging signals from multiple modalities, especially images and point clouds, for 3D perception tasks. Frustum PointNet (Qi et al., 2018) uses 2D detection

on images to generate bounding boxes, which further guide the 3D detection with point clouds. Ku et al. (2018); Chen et al. (2017); Liang et al. (2019) project point clouds to BEV frame, and then fuse 2D RGB features with BEV features to generate proposals and regress bounding boxes. Alternatively, Pointpainting (Vora et al., 2020) proposes to augment point clouds with image semantic segmentation results to leverage color information. Recent work (Zhu et al., 2021; Yoo et al., 2020) starts to explore deep feature-level fusion between points and images. CLOCs (Pang et al., 2020) proposes a post-detection fusion mechanism to combine the candidate boxes from RGB and LiDAR inputs. Ye et al. (2020) propose to use feature alignment between point clouds and images to improve monocular 3D object detection. Under the umbrella of the cross-modality setting, 2DPASS (Yan et al., 2022) transfers features learned from images to the Lidar during training to help the Lidar model to perform 3D semantic segmentation during inference. BEVDepth (Li et al., 2022) proposes to obtain reliable depth estimation for 3D object detection by exploiting the camera parameters together with the image features during training. On the contrary, our method explores a more realistic yet challenging setting, where we use Lidar during training to help the camera-only model in 3D perception. Despite outperforming single-modality methods, most prior work assumes identical domain distribution during training and inference, which is not always true in reality. As a result, the actual usefulness of additional sensors is still unclear when domain shift exists.

## 2.2 CROSS-DOMAIN 3D PERCEPTION

While extensive research has been conducted on domain adaptation for 2D tasks, the field of domain adaptation for 3D perception in the real world has relatively small literature. Some prior work adapts depth estimation from synthetic to real image domains (Kundu et al., 2018; Zhao et al., 2019). Working on point clouds, Qin et al. (2019) design a multi-scale adaptation model for 3D classification. For 3D semantic segmentation, Wu et al. (2019) project the point clouds to 2D view, while some other work (Jaritz et al., 2020; Peng et al., 2021; Gong et al., 2021) proposes to leverage point clouds and images data together. Recent work (Zhang et al., 2021; Yang et al., 2021; Luo et al., 2021) starts to explore cross-domain 3D object detection from point clouds. Tarvainen & Valpola (2017) employ Mean Teacher to generate pseudo-label for the target domain. SRDAN (Zhang et al., 2021) employs adversarial learning to align the features between different domains. Although prior work (Jaritz et al., 2020; Li et al., 2021) explores various domain adaptation techniques for different sensor modalities, these methods only adopt the same modalities to learn the domain shift between source and target data. In contrast, our approach achieves robust 3D perception in a more general scenario, where the model can perform accurate 3D inference in the target domain by adapting information encoded in source-exclusive modalities.

## 2.3 3D INFERENCE IN BIRD'S-EYE-VIEW FRAME

Inferring 3D scenes from the BEV perspective has recently received a large amount of interest due to its effectiveness. MonoLayout (Mani et al., 2020) estimates the layout of urban driving scenes from images in the BEV frame and uses an adversarial loss to enhance the learning of hidden objects. Can et al. (2021) propose to employ graphical representation and temporal aggregation for better inference in the driving scenarios using on-board cameras. Recently, using BEV representation to merge images from multiple camera sensors has become a popular approach (Hendy et al., 2020; Pan et al., 2020). Following the monocular feature projection proposed by Orthographic Feature Transform (OFT) (Roddick et al., 2018), Lift-Splat-Shoot (Philion & Fidler, 2020) disentangles feature learning and depth inference by learning a depth distribution over pixels to convert camera image features into BEV. Unlike the above work performing BEV analysis in settings with more controlled premises, we are the first to explore cross-domain and cross-sensor settings, leading to a more robust and more realistic 3D inference methodology.

## 3 APPROACH

In this work, we consider the task of learning BEV representation of scenes with domain shift and modality mismatch. Specifically, the model will be given annotated LiDAR point clouds and cameras images in the source domain, but only unannotated camera images in the target domain. And the model seeks to achieve highest performance on the unsupervised target domain. This problem setting is common and worthwhile, especially considering the existence of many existing public multi-modality datasets and the rise of many camera-only vehicle scenarios.

Formally, **for the source domain**, we are given *labeled* data with $N^s$ multi-modality samples, $\mathcal{D}^s = \{(\boldsymbol{X}_i^s, \boldsymbol{P}_i^s, \boldsymbol{y}_i^s)\}_{i=1}^{N^s}$, where $s$ represents the source domain. Here $\boldsymbol{X}_i^s = \{\boldsymbol{x}_k^s\}_{k=1}^n$ consists of

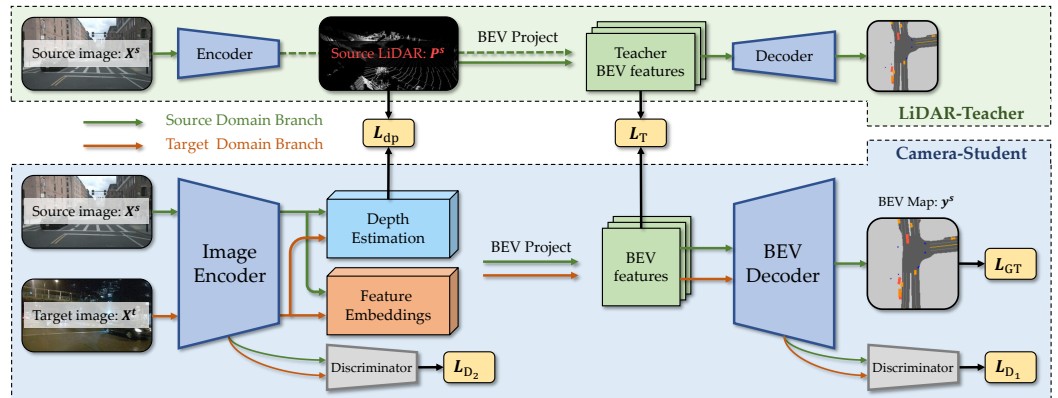

Figure 2: **Overview of our CroMA framework**. CroMA includes three components. (1) **LiDAR-Teacher** uses voxelized LiDAR point clouds to transform the image features to BEV frame. It provides essential knowledge on how to guide image learning given LiDAR information. (2) **Camera-Student** leverages the same architecture as the teacher, but the depth is estimated. It is supervised by teacher model as well as the LiDAR ground truth. (3) **Discriminators** are used to align features from source and target domains.

$n$ camera images $\boldsymbol{x}_k^s \in \mathbb{R}^{3 \times H \times W}$. The number of cameras $n$ can take any integer as small as one, depending on the dataset used or cameras deployed on the vehicle. In addition, each camera image has an intrinsic matrix and an extrinsic matrix. $\boldsymbol{P}_i^s$ is a point cloud containing multiple unordered points $\boldsymbol{p} \in \mathbb{R}^3$ represented by 3D coordinate values. And label $\boldsymbol{y}_i^s$ represents rasterized representation of the scenes in the BEV coordinate. **For the target domain**, we are given *unlabeled* data with $N^t$ single-modality (image) samples, $\mathcal{D}^t = \{\boldsymbol{X}_i^t\}_{i=1}^{N^t}$, where $t$ represents the target domain, and we would like to estimate $\{\boldsymbol{y}_i^t\}_{i=1}^{N^t}$, the BEV representation of the scenes in the target domain.

An overview of our method CroMA is illustrated in Figure 2. CroMA is designed to extract features from monocular images and project the features into BEV frame (Section 3.1), using estimated or ground truth 3D depth information. The model is composed of a LiDAR-teacher and a Camera-student (Section 3.2), where the teacher encodes how to learn better representation given LiDAR point clouds, and transfers that knowledge to the camera-only student using multi-level teacher-student supervision. Finally, to bridge the domain gap between source and target domains, we leverage adversarial discriminators at different feature layers to align the distributions across two domains in the camera-student model (Section 3.3). Finally, we describe the overall learning objective and loss designs (Section 3.4).

## 3.1 LEARNING BEV FROM IMAGES

In order to achieve 3D perception under the cross-modality setting, our first challenge is to unify the image coordinates, point cloud coordinates, and BEV coordinates into a joint space. We follow LSS (Philion & Fidler, 2020) to transform the image features from its perspective view into the BEV view. More specifically, we tackle this problem by constructing a 3D voxel representation of the scene for each input image. We discretize the depth axis into $N_d$ bins and lift each pixel of the images into multiple voxels (frustums), where each voxel is represented by the 3D coordinate of its center location. For a given pixel $\text{px} = (h, w)$ on one of the camera image, it corresponds to a set of $N_d$ voxels at different depth bins.

$$V_{\text{px}} = \{v_i = M^{-1}[d_i h, d_i w, d_i]^T | i \in \{1, 2, \cdots, N_d\}\}, \tag{1}$$

where $M$ is camera matrix and $d_i$ is the depth of the $i$-th depth bin. The feature vector of each voxel $v_i$ in $V_{\text{px}}$ is the base feature $\boldsymbol{f}_{\text{px}}$ of pixel px scaled by the depth value $\alpha_i$. More specifically, $\boldsymbol{f}_{v_i \in V_{\text{px}}} = \alpha_i \cdot \boldsymbol{f}_{\text{px}}$, where the pixel feature $\boldsymbol{f}_{\text{px}}$ is extracted by an image encoder. And the depth value $\alpha_i$ is obtained either from LiDAR point clouds or by estimation, in the teacher and student model, respectively. The acquirement of $\alpha_i$ is introduced in Sec. 3.2.

After getting the features for each of the voxels, we project the voxels onto the BEV and aggregate the features to get the BEV feature map. The BEV frame is rasterized into $(X, Y)$ 2D grids, and for each grid, its feature is constructed from the features of all the 3D voxels projected into it using mean pooling. This projection allows us to transform arbitrary number of camera images into a unified

BEV frame. Finally, we obtain an image-like BEV feature embedding, which is used to estimate the final representation using a convolutional neural network (CNN) decoder.

This architecture design bridges the image and LiDAR modalities through an intermediate 3D voxelized representation. Hence, we can input LiDAR point clouds into the model to directly guide the BEV projection without having to change the overall pipeline. This further enables the distillation of knowledge from the point clouds to images using a teacher-student model.

### 3.2 CROSS-MODALITY TRANSFER WITH TEACHER-STUDENT DISTILLATION

The co-existence of domain and modality gaps poses additional challenges to the adaptation task. Although the Lidar sensor in the source domain provides 3D knowledge to the model, it also increases the domain discrepancy between the source and the target, which could hurt the model adaptation (as we will see in Sec. 4.3 and Table 5) [1]. Hence, the unique difficulty of our work lies in exploiting the *Lidar point clouds* during training to guide the camera model for better 3D estimation.

**Depth Supervision by Point Clouds.** The main advantage of point clouds over the image modality is the accurate 3D positional information coming from the depth measurement. Due to the lack of LiDAR during evaluation, we cannot use point clouds as direct input of the model. Hence, one alternative approach to use point clouds is to supervise the depth estimation in the model. As in Eq. 1, for each pixel, our model calculates the feature of its corresponding voxels by multiplying the pixel voxel with a depth value $\alpha_i$. We use another depth head to predict a depth distribution $\boldsymbol{\alpha}_{\mathrm{px}} = \{\alpha_1, \alpha_2, \cdots, \alpha_{N_d}\}$ over $N_d$ depth bins for each pixel px.

The ground truth depth supervision for this estimation task is generated by LiDAR point clouds as follows: When projected to the image frame, the points corresponding to one pixel can have three conditions. If the pixel has, **(1) no point inside**: the ground truth depth distribution of it is omitted. **(2) only one point inside**: the ground truth depth distribution of this point is a one-hot vector, with value one being in the voxel that the point lies in. **(3) multiple points inside**: the ground truth depth distribution $\alpha_i$ of this point is calculated by counting the number of points in each depth bin, and dividing them by the total number of points: $\alpha_i = \frac{\text{Number of points in depth bin } v_i}{\text{Total number of points in } V_{\mathrm{px}}}$.

Using a distribution-based depth representation effectively accounts for the ambiguity when objects of different depth occur in one pixel. This happens at the boundary of the objects, and becomes more severe when images get downsampled and pixels become large during feature encoding. Moreover, a probabilistic depth representation considers uncertainty during depth estimation, and degenerates to pseudo-LiDAR methods (Weng & Kitani, 2019) if the one-hot constraint is added.

**Learning from LiDAR-Teacher.** Despite being intuitive and straightforward, direct depth supervision is not optimal for two reasons. First, LiDAR supervision is only on the intermediate feature layer, providing no supervision on the second half of the model. Also, while LiDAR provides accurate depth measurement, "depth estimation" is still different from our overall objective on BEV representation. Motivated by this, as shown in Figure 2, we propose to use a pretrained LiDAR oracle model to supervise the image model at the final BEV feature embedding, such that the supervision of LiDAR is provided to the whole model, and is closer to the final objective. We call the model using ground truth point cloud information "LiDAR-Teacher," and the model to be supervised "Camera-Student." This boils down to a knowledge distillation problem where the 3D inference knowledge of the LiDAR-teacher is distilled to the camera-only student. Note that the classic problem of "better teacher, worse student" (Cho & Hariharan, 2019; Mirzadeh et al., 2020; Zhu & Wang, 2021) in knowledge distillation due to capacity mismatch does not exist in this model, because the LiDAR-Teacher and Camera-Student models in CroMA are almost identical.

Overall, this teacher-student mechanism allows the camera model to learn better 3D representation from the point clouds, achieving more complete LiDAR supervision at different stages, while still keeping the model image-centric for image-only inference.

### 3.3 CROSS-DOMAIN ADAPTATION WITH ADVERSARIAL FEATURE ALIGNMENT

Since the BEV annotations and the LiDAR ground truth are only available in the source data, the model will be heavily biased to the source distribution during teacher-student supervision. Hence, we bridge the target and source domains using adversarial training. Specifically, we place one

---

[1]We provide a more detailed analysis over this perspective in the Appendix.

discriminator $D_1$ at BEV decoder CNN blocks, and another $D_2$ at the image encoder CNN blocks, to align the features of two domains by optimizing over discriminator losses. While the final-layer discriminator $D_1$ is constantly useful to align features learned from the LiDAR-Teacher and final ground truth, we find that the middle-layer discriminator $D_2$ is exceptionally effective under certain domain gaps where images have great changes but LiDAR remains robust.

To achieve adversarial learning, given a feature encoder $E$ and input sample $X$, a domain discriminator $D$ is used to discriminate whether the feature $E(X)$ comes from the source domain or the target domain. The target and source domain samples are given the label $d = 1$ and $d = 0$, respectively. And $D(E(X))$ outputs the probability of the sample $X$ belonging to the target domain. Hence, the discriminator loss is formulated by a cross-entropy loss:

$$\mathcal{L}_{\mathrm{dis}} = d \log D(E(X)) + (1 - d) \log(1 - D(E(X))). \tag{2}$$

Moreover, in order to learn domain-invariant features, our feature encoder $E$ should try to extract features that fool the discriminator $D$, while the discriminator $D$ tries to distinguish the right domain label of the samples. This adversarial strategy can be formulated as a "min-max" optimization problem: $\mathcal{L}_{\mathrm{D}} = \min_E \max_D \mathcal{L}_{\mathrm{dis}}$. The "min-max" problem is achieved by a Gradient Reverse Layer (GRL) (Ganin & Lempitsky, 2015), which produces reverse gradient from the discriminator $D$ to learn the domain-invariant encoder $E$. The loss form is the same for both $D_1$ and $D_2$.

### 3.4 Full Objective and Inference

The overall objective of our model is composed of the supervision from the BEV ground truth, the LiDAR-Teacher, and the domain alignment discriminators. Given the output rasterized BEV representation map $\boldsymbol{y} \in \mathbb{R}^{X \times Y \times C}$, the ground truth (GT) loss term $\mathcal{L}_{\mathrm{GT}}$ can be formulated as a cross-entropy loss between the estimated source domain BEV map $\tilde{\boldsymbol{y}}^s$ and the GT label $\boldsymbol{y}^s$:

$$\mathcal{L}_{\mathrm{GT}}(\tilde{\boldsymbol{y}}^s, \boldsymbol{y}^s) = -\sum_{i=1}^{X} \sum_{j=i}^{Y} \sum_{k=1}^{C} y_{(i,j,k)}^s \log \tilde{y}_{(i,j,k)}^s, \tag{3}$$

The supervision from the LiDAR-Teacher is composed of a direct depth estimation loss $\mathcal{L}_{\mathrm{dp}}$ and a teacher feature supervision $\mathcal{L}_{\mathrm{T}}$. As described in Sec. 3.1, given the 3D depth volume $\boldsymbol{\alpha} \in \mathbb{R}^{H \times W \times N_d}$, the direct depth supervision term $\mathcal{L}_{\mathrm{dp}}$ is formulated as a cross entropy loss between the estimated 3D depth distribution volume $\tilde{\boldsymbol{\alpha}}^s$ in the source domain, and the GT depth volume $\boldsymbol{\alpha}^s$ calculated from LiDAR point clouds as described in Sec. 3.2:

$$\mathcal{L}_{\mathrm{dp}}(\tilde{\boldsymbol{\alpha}}^s, \boldsymbol{\alpha}^s) = -\sum_{i=1}^{H} \sum_{j=i}^{W} \sum_{k=1}^{N_d} \alpha_{(i,j,k)}^s \log \tilde{\alpha}_{(i,j,k)}^s, \tag{4}$$

And for the LiDAR-Teacher feature supervision: $\mathcal{L}_{\mathrm{T}}(\boldsymbol{F}^{\mathrm{te}}, \boldsymbol{F}^{\mathrm{st}}) = \mathcal{L}_2(\boldsymbol{F}^{\mathrm{te}}, \boldsymbol{F}^{\mathrm{st}})$ is an $\mathcal{L}_2$ loss, where $\boldsymbol{F}^{\mathrm{te}}$ and $\boldsymbol{F}^{\mathrm{st}}$ are the feature maps of teacher and student models, respectively. Finally, the domain adaptation loss contains $\mathcal{L}_{\mathrm{D}_1}$ and $\mathcal{L}_{\mathrm{D}_2}$ with the form described in Eq. 2.

**The final objective** is formulated as a multi-task optimization problem:

$$\mathcal{L}_{\mathrm{CroMA}} = \mathcal{L}_{\mathrm{F}} + \lambda_{\mathrm{T}} \mathcal{L}_{\mathrm{T}} + \lambda_{\mathrm{dp}} \mathcal{L}_{\mathrm{dp}} + \lambda_{\mathrm{D}_1} \mathcal{L}_{\mathrm{D}_1} + \lambda_{\mathrm{D}_2} \mathcal{L}_{\mathrm{D}_2}, \tag{5}$$

where $\lambda_{\mathrm{T}}, \lambda_{\mathrm{dp}}, \lambda_{\mathrm{D}_1}$, and $\lambda_{\mathrm{D}_2}$ are weights for the corresponding loss terms. The CroMA model is trained end-to-end using the loss term in Eq. 5. During inference, target samples go into the Camera-Student model to output the final BEV representation. More training details are provided in Sec. 4.

## 4 Experiments

### 4.1 Datasets and Domain Settings

We evaluate CroMA with four unique domain shift settings, constructed from two large-scale datasets, nuScenes (Caesar et al., 2020) and Lyft (Kesten et al., 2019). The first three domain settings are subsampled from the nuScenes dataset, including *city-to-city*, *day-to-night*, and *dry-to-rain* adaptation. In addition, we also perform another inter-dataset domain transfer: *dataset-to-dataset* adaptation, where the source and target domain data are from nuScenes and Lyft datasets. All adaptation settings follow the assumption that the source has access to cameras and Lidar sensors, while the target only has cameras. We use all six cameras provided by the nuScenes dataset. We also analyze surprising observations on cross-modality performance in the ablation study. More details regarding the domain settings and dataset splits are in the Appendix Sec. B.

Table 1: CroMA achieves best performance on all classes under ***city-to-city*** domain gaps in IoU. ***DA*** and ***CM*** denote whether a model considers domain adaptation and cross-modality in design.

| Boston → Singapore | DA | CM | Vehicle | Road | Lane | Singapore. → Boston | DA | CM | Vehicle | Road | Lane |
|---|---|---|---|---|---|---|---|---|---|---|---|
| MonoLayout (Mani et al., 2020) | ✗ | ✗ | 14.2 | 35.9 | 7.5 | MonoLayout (Mani et al., 2020) | ✗ | ✗ | 14.4 | 38.7 | 8.4 |
| OFT (Roddick et al., 2018) | ✗ | ✗ | 16.8 | 37.9 | 9.6 | OFT (Roddick et al., 2018) | ✗ | ✗ | 16.1 | 41.8 | 10.0 |
| LSS (Philion & Fidler, 2020) | ✗ | ✗ | 17.6 | 38.2 | 10.6 | LSS (Philion & Fidler, 2020) | ✗ | ✗ | 19.5 | 42.3 | 10.7 |
| Wide-range Aug. | ✓ | ✗ | 17.9 | 40.5 | 12.4 | Wide-range Aug. | ✓ | ✗ | 19.9 | 43.7 | 11.5 |
| Vanilla DA | ✓ | ✗ | 13.0 | 31.4 | 9.1 | Vanilla DA | ✓ | ✗ | 21.2 | 45.7 | 12.5 |
| Depth-Supv DA | ✓ | ✓ | 19.0 | 42.8 | 14.9 | Depth-Supv DA | ✓ | ✓ | 22.5 | 47.1 | 12.9 |
| Input-fusion Teacher | ✓ | ✓ | 18.6 | 42.7 | 14.1 | Input-fusion Teacher | ✓ | ✓ | 22.0 | 47.5 | 13.1 |
| **CroMA (ours)** | ✓ | ✓ | **20.5** | **43.1** | **15.6** | **CroMA (ours)** | ✓ | ✓ | **25.4** | **48.9** | **14.4** |

Table 2: CroMA leads to significant improvements under ***day-to-night*** domain shift, and also achieves best results under ***dry-to-rain*** domain shift in IoU.

| Day → Night | DA | CM | Vehicle | Road | Lane | Dry → Rain | DA | CM | Vehicle | Road | Lane |
|---|---|---|---|---|---|---|---|---|---|---|---|
| MonoLayout Mani et al. (2020) | ✗ | ✗ | 5.9 | 37.7 | 5.9 | MonoLayout (Mani et al., 2020) | ✗ | ✗ | 20.6 | 68.7 | 13.1 |
| OFT (Roddick et al., 2018) | ✗ | ✗ | 6.6 | 40.5 | 6.0 | OFT (Roddick et al., 2018) | ✗ | ✗ | 24.1 | 79.8 | 16.2 |
| LSS (Philion & Fidler, 2020) | ✗ | ✗ | 6.7 | 41.2 | 7.1 | LSS (Philion & Fidler, 2020) | ✗ | ✗ | 27.8 | 71.0 | 16.8 |
| Wide-range Aug. | ✓ | ✗ | 10.3 | 46.0 | 10.4 | Wide-range Aug. | ✓ | ✗ | 28.2 | 71.2 | 17.2 |
| Vanilla DA | ✓ | ✗ | 11.2 | 48.8 | 11.1 | Vanilla DA | ✓ | ✗ | 29.1 | 70.8 | 18.3 |
| Depth-Supv DA | ✓ | ✓ | 15.7 | 50.5 | 14.2 | Depth-Supv DA | ✓ | ✓ | **29.6** | 71.8 | 19.1 |
| Input-fusion Teacher | ✓ | ✓ | 14.9 | 48.8 | 13.1 | Input-fusion Teacher | ✓ | ✓ | 29.5 | 71.0 | 18.8 |
| **CroMA (ours)** | ✓ | ✓ | **17.0** | **51.8** | **16.9** | **CroMA (ours)** | ✓ | ✓ | **29.6** | **71.9** | **19.5** |

## 4.2 Results and Comparisons

**Baselines** We compare our method with state-of-the-art BEV 3D layout perception work MonoLayout (Mani et al., 2020), OFT (Roddick et al., 2018), LSS (Philion & Fidler, 2020), as well as other baseline methods in domain adaptation and cross-modality learning. *Wide-range Aug.* means using a wide range of random scaling augmentation which potentially includes the target domain scale. For *Vanilla DA*, we adapt camera-only DA-Faster (Chen et al., 2018) to our BEV perception setting. *Depth-Supv DA* stands for depth supervised domain adaptation. We use source domain LiDAR as ground truth to supervise the depth estimation during training, without LiDAR-Teacher supervision (only $\mathcal{L}_{dp}$ without $\mathcal{L}_T$). *Input-fusion Teacher* is an alternative way of designing the LiDAR-Teacher, where we directly fuse point $(x, y, z)$ coordinates into their corresponding image pixels as additional channels in the teacher model, similar to Pointpainting (Vora et al., 2020). We use ***DA*** and ***CM*** to denote whether a model considers domain adaptation and cross-modality in design. Results are reported on vehicle, drivable roads, and lane marking classes using intersection-over-union (IoU).

**City-to-City Adaptation** As shown in Table 1, we observe that our CroMA model achieves the best performance on all classes for two inter-city transfer settings. Without domain adaptation, baseline approaches MonoLayout, OFT, and LSS all suffer from performance degradation. By considering the visual domain shift, wide-range augmentation and vanilla domain adaptation methods improve the baseline up to 11.2. Direct depth supervision and alternative input-fusion teacher models do not bring as much improvement as CroMA. The results clearly demonstrate the effectiveness of our method by distilling and aligning the LiDAR information for cross-modality domain adaptation 3D BEV perception.

**Day-to-Night Adaptation** As shown in Table 2 on the left, we observe that our CroMA model achieves the best performance in all classes. Different from the previous setting, we notice that the improvement on Day → Night setting is exceptionally high. This is because the initial domain gap between day and night scenarios is very large in the camera modality space. Moreover, the LiDAR sensor is robust under illumination changes due to its active imaging mechanism as opposed to camera's passive one. Thus, incorporating LiDAR point cloud information helps the model to learn a more robust illumination-invariant representation from the image inputs.

**Dry-to-Rain Adaptation** As shown in Table 2 on the right, under this setting we also observe that our CroMA model achieves the best performance on all classes. However, we notice that the improvement under Dry → Rain setting is not as big as previous settings, and is only on par with other alternative cross-modality baselines. This is because the domain gap between dry and rain scenarios is not big in image modality. Hence, baseline methods OFT and LSS are already able to obtain decent results even without domain adaptation. Furthermore, rainy weather is known to cause great domain shift in the LiDAR modality (Xu et al., 2021). As a result, the knowledge learned from source-exclusive LiDAR suffers from an unknown domain shift which can be larger than the image modality domain shift. This can potentially cancel out the benefit of 3D information learned from point clouds and explains for the smaller improvement.

Table 3: Our model achieves the best performance under *dataset-to-dataset* domain gaps.

| nuScenes → Lyft | DA | CM | Vehicle |
|---|---|---|---|
| MonoLayout (Mani et al., 2020) | ✗ | ✗ | 11.8 |
| OFT (Roddick et al., 2018) | ✗ | ✗ | 16.5 |
| LSS (Philion & Fidler, 2020) | ✗ | ✗ | 19.9 |
| Wide-range Aug. | ✓* | ✗ | 21.9 |
| Vanilla DA | ✓ | ✗ | 22.5 |
| Depth-Supv DA | ✓ | ✓ | 23.4 |
| Input-fusion Teacher | ✓ | ✓ | 22.8 |
| **CroMA (ours)** | ✓ | ✓ | **24.4** |

Table 4: Our proposed LiDAR-Teacher also leads to the best performance without domain shift, which demonstrate the robustness of our CroMA model.

| w/o domain shift | CM | Vehicle | Road | Lane |
|---|---|---|---|---|
| MonoLayout (Mani et al., 2020) | ✗ | 24.3 | 69.0 | 13.1 |
| FishingNet (Hendy et al., 2020) | ✗ | 30.0 | - | - |
| OFT (Roddick et al., 2018) | ✗ | 30.1 | 72.2 | 16.9 |
| LSS (Philion & Fidler, 2020) | ✗ | 32.1 | 74.1 | 18.8 |
| Depth-Supv | ✓ | 34.8 | 75.8 | 19.1 |
| Input-fusion Teacher | ✓ | 35.1 | 76.5 | 18.7 |
| **CroMA (ours)** | ✓ | **35.8** | **76.8** | **20.5** |

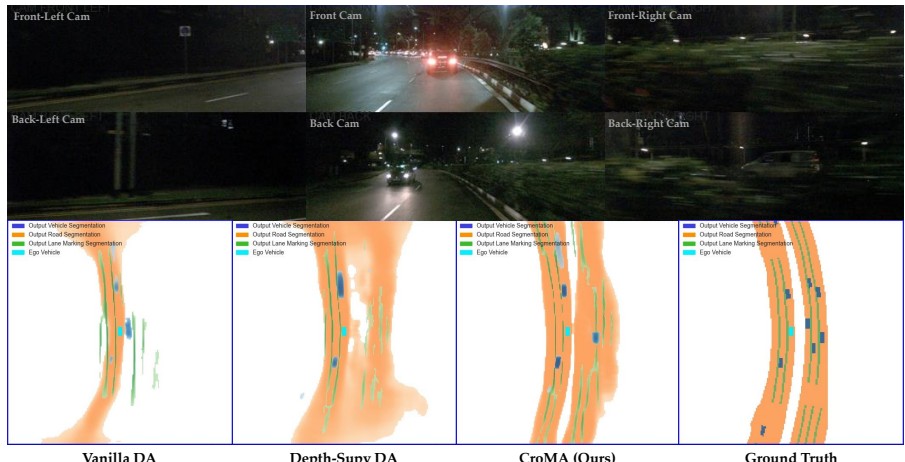

Figure 3: **Qualitative Results in Day → Night setting** (model is trained with daytime data, and validated with night data). We notice that CroMA performs significantly better than other baselines for vehicles, drivable roads, and lane marking classes. From **left** to **right**: (1) Vanilla adversarial learning; (2) LiDAR as depth supervision with adversarial learning; (3) our CroMA model; (4) Ground Truth. Best viewed in color.

**Dataset-to-Dataset Adaptation** As shown in Table 3, we can also observe that our CroMA model achieves best performance in nuScenes → Lyft setting. Following Philion & Fidler (2020), because Lyft does not provide road segment and lane marking information in the HD map, we report results on the vehicle class. Note that there is not pre-defined train and validation data split for Lyft, so we choose a new split and report the number of our re-implemented results for the baselines. Compared with baselines with and without domain adaptation or cross-modality learning, our CroMA model does a better job in leveraging and adapting LiDAR information. We include Lyft → nuScenes results in the Appendix Sec. C.

**Qualitative Results** As shown in Figure 3, under Day → Night domain shift setting, our model achieves significantly better monocular 3D perception than the vanilla DA and Depth-Supv DA baselines. We observe that CroMA provides more clearly defined road boundaries and lane markings. The depth and size of the vehicles and the road on the right side are also predicted more accurately. CroMA only misses some vehicles that are hardly visible in camera due to occlusion and distance. Overall, the qualitative results validate the effectiveness of CroMA in closing the gap between data domains and leveraging point clouds information for better 3D inference.

## 4.3 Analysis over Surprising Observations and Ablation Study

**Naive Usage of More Sensors Leads to Worse Performance** It is naturally believed that introducing multiple sensors in the perception model is bound to increase the model performance. Surprisingly, experiments shown in Table 5 negates this naive intuition. When we introduce LiDAR sensor in the source domain as depth supervision, the result decreases for 0.3. As we described in Sec. 3.2, the domain distribution divergence increases after introducing the sensor-modality shift. As a result, we propose multiple components in CroMA to account for the visual and sensor domain shift. Experiments show that while wide augmentation strategy and adversarial discriminator both achieve better results than baseline (11.2 vs. 6.7 in IoU), our LiDAR-Teacher further boosts the result to 17.0 by leveraging effective LiDAR knowledge distillation and alignment.

Table 5: Ablation study shows that our proposed components all contribute to the final state-of-the-art performance. We report results on vehicle class under ***day-to-night*** domain gap in IoU.

| Backbone | Wide Augmentation | Adversarial Discriminators | LiDAR Supervision | LiDAR-Teacher | Results |
|:---:|:---:|:---:|:---:|:---:|:---|
| ✓ | | | | | 6.7 |
| ✓ | | | ✓ | | 6.4 (**-0.3**) |
| ✓ | ✓ | | | | 10.3 (**+3.6**) |
| ✓ | ✓ | ✓ | | | 11.2 (**+4.5**) |
| ✓ | ✓ | ✓ | ✓ | | 15.7 (**+9.0**) |
| ✓ | ✓ | ✓ | ✓ | ✓ | **17.0** (**+10.3**) |

Figure 4: The proposed progressive learning strategy effectively addresses the challenge caused by mixed domain gap scenario (***Boston-to-Singapore*** mixed with ***day-to-night***) on nuScenes. This shows that CroMA can function in a more realistic domain shift setting.

| **Mixed Domain Gap** | Vehicle | Road | Lane |
|:---|:---:|:---:|:---:|
| Direct Inference | 17.6 | 38.2 | 10.6 |
| Vanilla DA | 13.0 | 31.4 | 9.1 |
| Progressive DA | 18.8 | 41.5 | 13.2 |
| **CroMA (ours)** | **20.5** | **43.1** | **15.6** |

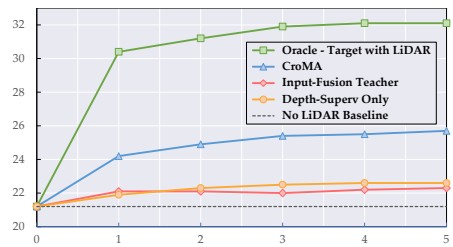

Figure 5: Results of CroMA improve as the number of LiDAR sweeps increases.

**Dealing with Mixed Domain Shift** Another common but underexplored question we observe in the 3D domain adaption setting is the mixed domain shift problem, where multiple types of gaps between source and target domains often occur concurrently. For example, in the nuScenes dataset, the Boston data is only captured during the day, while the Singapore data is captured both day and night. This leads to the fact that some of the data in the Singapore domain only has a city-wise gap to the source, while the other part data has both city-wise and illumination-wise gaps. As shown in Table 4, we find that directly leveraging adversarial discriminator in this scenario leads to worse performance than direct inference (model trained without domain adaptation), because mixed domain in the target domain confuses the domain discriminator. Hence, we propose a progressive learning mechanism, where we first perform adaptation with city-wise data for $100K$ steps, and then train the model on the full target domain dataset for another $150K$ steps. This effectively alleviates the mixed domain shift problem, and helps CroMA achieve leading results than other baselines.

**Results within Same Domain** In addition to the domain shift setting, we further validate that our proposed LiDAR-Teacher is able to distill knowledge and achieve the best performance within the same domain. As shown in Table 4, the results are obtained in the original nuScenes train/val split.

**Effect on LiDAR Density & Comparison with Oracle Model** As shown in Figure 5, we validate that our model can achieve higher performance when denser LiDAR is available. This can be achieved by grouping continuous scans of LiDAR point clouds (from 1 to 5) into a whole, to have a denser 3D representation of the scene. We observe that other cross-modality baselines including *Input-Fusion Teacher* and *Depth-Supv* models cannot effectively leverage the LiDAR knowledge, even with dense point clouds available. We also compare our model with the LiDAR oracle model (target domain also has LiDAR modality) and find that the gap between the upper bound result and No-LiDAR baseline is significantly reduced. The remaining performance gap is caused by the unknown LiDAR domain gap which we hope to further reduce in future work.

**More results in Sec. C** We also find that the different types of domain gaps react unevenly to different model designs, and conduct more ablation on model designs, including depth ground truth signals and LiDAR-Teacher. These can be found in the Appendix.

## 5 CONCLUSION

In this paper, we proposed CroMA to estimate 3D scene representation in BEV under domain shift and modality change. To achieve this, we construct a LiDAR-Teacher and distill knowledge from it into an Camera-Student by feature supervision. And we further propose to align feature space between the domains using multi-stage adversarial learning. Results on large-scale datasets with various challenging domain gaps demonstrated the effectiveness of our approach, which marks a significant step towards robust 3D scene perception in the wild.

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

# APPENDIX

## A  VISUALIZATION WITH VIDEO CLIPS

For video visualization, please see the **ICLR-supp-video.mp4** in the supplementary files.

## B  DATASET AND IMPLEMENTATION DETAILS

### B.1  IMPLEMENTATION DETAILS

Following Philion & Fidler (2020), we use EfficientNet (Tan & Le, 2019) pretrained on ImageNet (Deng et al., 2009) as our image encoder backbone. Two heads are applied to estimate pixel features and pixel-wise depth distribution from the $8\times$ downsampled feature map. The 3D feature maps are projected to the bird's-eye-view frame using mean pooling. For the bird's-eye-view decoder we use ResNet-18 (He et al., 2016) as backbone, and upsample the features learned from the first three meta-layers of ResNet to the final BEV output. The $D_1$ and $D_2$ domain discriminators are applied to the output feature layers of EfficientNet and ResNet backbone, respectively. We use a light weight discriminator architecture, which is composed of a global averaging pooling layer, followed by two fully connected layers, and outputs the domain label. For input, we resize and crop input images to size $128 \times 352$. For output, we consider a $100$ meters $\times$ $100$ meters range centered at the ego-vehicle, with the grid size set to be $0.5$ meters $\times$ $0.5$ meters. The depth bin is set to be $1.0$ meter between $4.0$ meters and $45.0$ meters range. The whole model is trained end-to-end, with $\lambda_T = 1.0, \lambda_{dp} = 0.05, \lambda_{D_1} = 0.1, \lambda_{D_2} = 0.01$. We train CroMA using the Adam (Kingma & Ba, 2014) optimizer with learning rate $0.001$ and weight decay $1e$-7 for $50$K steps for the teacher model, and $200$K for the student model. We use horizontal flipping, random cropping, rotation, and color jittering augmentation during training. The whole model is implemented using the PyTorch framework (Paszke et al., 2019).

### B.2  DATASET DETAILS

For this section, we explain our dataset split in more details. For our experiments, we always split the target domain into two subsets for fair comparison. One of them can be accessed during training for adversarial learning and domain alignment, and the other is held out exclusively for validation.

We follow existing Lidar-based domain adaptation work, including SRDAN (Zhang et al., 2021), ST3D (Yang et al., 2021), UDA3D (Luo et al., 2021), and xMUDA (Jaritz et al., 2020), to create the domain split strategies. Specifically, for the *day-to-night*, *city-to-city*, and *dry-to-rain* settings, as described in prior approaches, we use the sentence in the nuScenes dataset and filter the keywords to split the dataset into corresponding subsets. For the *dataset-to-dataset* setting, we use the official split of the nuScenes dataset, and the split provided in ST3D (Yang et al., 2021) for the Lyft dataset. More details are provides as follows.

**City to City Adaptation**  For the *city-to-city* adaptation scenario, we sub-sample the trainval split of the large-scale dataset nuScenes (Caesar et al., 2020) captured in Boston and Singapore city. We treat one city as the source domain and the other as the target domain. The Boston part of data has 467 scenes in total, which is separated into 350 scenes for training and 117 scenes for validation. And the Singapore part of data has 383 scenes in total, which is separated into 287 scenes for training and 96 scenes for validation.

**Day to Night Adaptation**  For the *day-to-night* adaptation scenario, we also sub-sample the trainval split of the large-scale dataset nuScenes (Caesar et al., 2020). Every scene in the nuScenes dataset has a sentence of description, which can be parsed and used to categorize it into certain class. In this way, we create a day scene subset and a night scene subset out of the whole dataset. We treat day as the source domain and night as the target domain, because the night subset has significantly fewer samples than the day subset. The day split has 751 scenes which are all used for training. And the night part has 99 scenes in total, which is separated into 74 scenes for training and 25 scenes for validation.

Table 6: CroMA achieves great inference time compared with the baseline methods.

| | #Params (M) | Frame-per-Second (FPS) |
|---|---|---|
| OFT (Roddick et al., 2018) | 22 | 25 |
| LSS (Philion & Fidler, 2020) | 14 | 35 |
| **CroMA (Ours)** | 15 | 33 |

**Dry to Rain Adaptation**  For the *dry-to-rain* adaptation scenario, we also sub-sample the trainval split of the large-scale dataset nuScenes (Caesar et al., 2020). Similarly, using the scene description sentence, we create a dry (non-rainy) scene subset and a rainy scene subset out of the whole dataset. We treat dry as the source domain and rain as the target domain, because the rain subset has significantly fewer samples than the dry subset. The dry split has 685 scenes which are all used for training. And the rain part of data has 165 scenes in total, which is separated into 124 scenes for training and 41 scenes for validation.

**Dataset to Dataset Adaptation**  For the *dataset-to-dataset* adaptation scenario, we use the two large-scale autonomous driving datasets nuScenes (Caesar et al., 2020) and Lyft (Kesten et al., 2019). We treat one dataset as the source domain and the other as the target domain. For the nuScenes dataset, we use the original train and validation split, which has 700 scenes and 150 scenes, respectively. The Lyft dataset does not have an original split, so we sub-sample 132 scenes for training and 48 scenes for validation.

### B.3 Computational Complexity

Table 6 summarizes the number of parameters and inference speed for prior baselines and our model. We can see that our Lidar-Teacher distillation and multi-level adversarial learning modules do not affect the inference efficiency of CroMA compared with the baseline model. Our total number of parameters is 15M, and our inference time is 33 Frame-per-Second (FPS) on a V100 GPU, which is on par with the baseline LSS (Philion & Fidler, 2020). The training time for our model is around 20 hours on 4×V100 GPUs.

## C  Additional Results and Analysis

**Unique Challenge due to Problem Formulation**  We provide another perspective to understand the challenges brought by the co-existence of cross-modality and cross-domain gaps, and thus further motivating the design of our architecture. It is proved that in domain adaptation, the target domain error $\epsilon^t(h)$ can be upper-bounded by the inequality (Ben-David et al., 2010):

$$\epsilon^t(h) \leq \epsilon^s(h) + d_{\mathcal{H}\Delta\mathcal{H}}\left(P_X^s, P_X^t\right) + C, \tag{6}$$

where the bound is composed of the source-domain error $\epsilon^s(h)$, source-target distribution divergence $d_{\mathcal{H}\Delta\mathcal{H}}\left(P_X^s, P_X^t\right)$, and another term which is considered constant in our case. Existing domain adaptation work mostly focuses on reducing the source-domain error. However, while the introduction of LiDAR sensor **reduces the first term**, it **increases the second term**, because the source and target domains have an additional sensor-type gap. It is also demonstrated in Sec. 4 that naively using source domain Lidar can even hinder the target performance rather than improve it. Hence, the unique difficulty of our work lies in *leveraging Lidar to reduce the source-domain error $\epsilon^s(h)$, and in the meanwhile, preventing distribution divergence $d_{\mathcal{H}\Delta\mathcal{H}}\left(P_X^s, P_X^t\right)$ from increasing too much.*

**Different Depth GT Signal Generation Methods**  As shown in Table 7, we validate CroMA's design choice in depth ground truth signal generation (Eq. 2 in main file) by comparing with different alternative methods. Note that all the methods have the same output when there is no point or only one point projected to a pixel. The difference only comes from the behaviour when multiple points are projected to one pixel.

- *Majority Voting* means to generate a one-hot GT vector by assigning "1" to the depth bin with the most number of points inside, and random select one bin when more than one bin with the most number of points.

Table 7: We validate the design choices of CroMA by comparing with various depth GT generation methods and different choices of teacher supervision signals. Results show that depth supervision generated by Point Number Distribution (Eq. 2 in main file), and feature-level supervision from LiDAR-Teacher help the model achieve the best performance.

| Design choices in Day → Night Setting | | Vehicle | Road | Lane |
|---|---|---|---|---|
| Depth Ground Truth Signal (From Point Clouds) | No LiDAR | 11.2 | 48.8 | 11.1 |
| | Majority Voting | 14.1 | 50.0 | 13.7 |
| | Random Selection | 13.8 | 49.5 | 12.6 |
| | Softmax Point Number Distribution | 13.1 | 49.9 | 11.9 |
| | **Point Number Distribution (Ours)** | **17.0** | **51.8** | **16.9** |
| LiDAR-Teacher Supervision | No LiDAR-Teacher | 15.7 | 50.5 | 14.2 |
| | Soft-label Supervision | 15.2 | 50.7 | 13.9 |
| | **Feature Supervision (Ours)** | **17.0** | **51.8** | **16.9** |

Table 8: CroMA model achieves best performance on all classes under **Lyft-to-nuScenes** domain gaps in IoU. *DA* and *CM* means domain adaptation and cross modality.

| Lyft → nuScenes | DA | CM | Vehicle |
|---|---|---|---|
| MonoLayout (Mani et al., 2020) | ✗ | ✗ | 7.1 |
| OFT (Roddick et al., 2018) | ✗ | ✗ | 11.9 |
| LSS (Philion & Fidler, 2020) | ✗ | ✗ | 13.8 |
| Wide-range Aug. | ✓ | ✗ | 14.6 |
| Vanilla DA | ✓ | ✗ | 15.1 |
| Depth-Supv DA | ✓ | ✓ | 16.5 |
| Input-fusion Teacher | ✓ | ✓ | 15.1 |
| **CroMA (ours)** | ✓ | ✓ | **19.2** |

- *Random Selection* means to generate a one-hot GT vector by randomly selecting one of all the projected points, and assigning "1" to the depth bin in which the selected point lies.

- *Softmax Point Number Distribution* means to count the number of points inside every depth bin, and then use softmax to turn this vector into a distribution.

Finally, in CroMA we use direct *Point Number Distribution*, where we count the number of points inside every depth bin, and divide the vector by the total number of points to get a depth distribution. Results show that direct point number distribution outperforms other counterparts in ground truth generation. One possible reason is because CroMA downsamples the feature map $8\times$ when estimating the 3D depth volume. This makes it much more common for multiple points with different depth values to fall in the same pixel. By contrast, the other two methods, *Majority Voting* and *Random Selection*, will lose valuable information in these cases. On the other hand, softmax activate over-smooth the distribution, and also assign small values to zero bins (depth bins with no point inside). Hence, we find that preserving the native depth distribution of the points is a better way to supervise CroMA in 3D evaluation.

**Different LiDAR-Teacher Supervision Design** As shown in Table 7, we also validate CroMA's design choice a LiDAR-Teacher Supervision by comparing with different alternative methods. In addition to the feature-level supervision used in CroMA, another commonly used teacher supervision is the soft label output. Specifically, *Soft-label Supervision* means to use the class distribution output of the teacher model as the supervision of the student model, as opposed to the one-hot vector from the ground truth annotation. We find that feature-level supervision performs better than the soft-label supervision. One reason is because we use a small number of classes, which makes the supervision from the soft-label less informative. Moreover, because the teacher and student models in CroMA have almost identical architecture and capacity, enforcing the corresponding feature level similarity between the teacher and student model provides a stronger supervision than the soft-label output, without harming the model learning. As future work, we will explore whether soft-label may be more useful when the number of classes in the model is larger.

**Lyft to nuScenes Adaptation** Due to page limit, Table 3 in the main file only shows the results under the nuScenes → Lyft setting. Here we also present the results under the Lyft → nuScenes adaptation setting for completeness. As shown in Table 8, we can also observe that our CroMA model achieves the best performance. Like previous scenarios, baseline approach MonoLayout,

Table 9: Different types of domain gaps react unevenly to different model designs. Direct depth supervision and the middle layer feature alignment block provide a larger improvement under Day → Night setting than Dry → Rain setting.

| CroMA Designs | Day → Night | Dry → Rain |
|---|---|---|
| Image-only Baseline | 11.2 | 28.3 |
| LiDAR Teacher Feature Supervision | 14.9  *(+3.7)* | 29.5  *(+1.2)* |
| LiDAR Teacher Feature + Depth Supervision | 17.0  *(+5.8)* | 29.6  *(+1.3)* |
| Without Domain Alignment | 7.1 | 28.1 |
| Feature Alignment at Final layer | 12.2  *(+5.1)* | 29.6  *(+1.5)* |
| Feature Alignment at Mid + Final layer | 17.0  *(+9.9)* | 29.3  *(+1.2)* |

Table 10: CroMA model achieves great performance with scaling augmentation and FoV matching for *nuScenes-to-Lyft* domain gaps in vehicle IoU.

| nuScenes → Lyft | Scaling Augmentation | Match FoV | Vehicle IoU |
|---|---|---|---|
| None | | | 23.5 |
| with scaling augmentation | ✓ | | 23.8 |
| with both | ✓ | ✓ | 24.4 |

OFT, and LSS without domain adaptation suffer from performance degradation due to domain shift. And compared with other baselines with domain adaptation or cross-modality learning, our CroMA does a better job in leveraging and adapting LiDAR information.

**Effect of Different Designs on Different Domain Gaps**  We find in our experiments that different types of domain gaps react unevenly to different model designs. As shown in Table 9, for Day → Night setting which has less domain shift in LiDAR point cloud modality than camera modality, the depth supervision, and the middle layer feature alignment block provide a large improvement on top of other modules. As opposed to Dry → Rain setting, where domain gap is larger for point clouds than images, and no significant improvement is achieved by using these two components. This further validates the necessity of using multiple modalities under domain adaptation settings, which can effectively improve the algorithm robustness under different domain shifts.

**Solving Scale Problem**  Scaling ambiguity is an inherent problem for monocular depth estimation. We solve this problem by using the random cropping, scaling augmentation strategy during training, and also by matching the FoV (Field-of-View) of two domains using their intrinsic matrices. The augmentation increases the robustness of the depth prediction model in scale difference. And the FoV matching scales the images in the target domain to match the FoV of the source domain. This makes sure that one object looks approximately the same size in images, if it is of the same distance to the ego vehicle in source and target domains, thus reducing the scale ambiguity. In Table 10, we provide an ablation study of scale augmentation and FoV matching in nuScenes-to-Lyft adaptation. As we can see, they both improve our final model performance.

From Table 10, we also notice that even without the two methods mentioned above, the model still performs fairly well, compared with the baseline methods in Table 3. This is because in driving scenarios, the camera FoV, the context in the images and the depth distribution of the images have a relatively strong prior – they do not have a strong discrepancy even across different driving scenarios (domains). This is different to more general depth estimation scenarios, where objects can have drastically different depth distribution and the intrinsic matrix can have big differences from image to image.

**Ablation in Semantic Segmentation and Depth Prediction**  From the task perspective, there are two domain gaps in this task, one in the semantic segmentation task and the other in the depth prediction task. In Table 11, we show the result of depth estimation and semantic segmentation alone in the Day to Night scenario.

For depth estimation, we report the cross entropy error (CEE) with the ground truth we described in Sec 3.2, because each pixel will have multiple depth values. For this task the Lidar teacher supervision refers to $L_{dp}$ in our main pipeline. We observe that the method we propose significantly improves the depth estimation metrics by 44.9%.

Table 11: Our proposed modules achieve great performance in both depth estimation and semantic segmentation tasks.

| Day → Night | Depth Estimation (CEE) | Semantic Segmentation (IoU) |
|---|---|---|
| Direct Inference | 2.56 | 27.5 |
| Adversarial Learning (AL) | 1.97 | 31.8 |
| Lidar-Teacher Supervision + AL | **1.41** | **32.1** |

Table 12: Naïve LiDAR supervision cannot help the final perception under cross-domain and cross-modality scenario.

| Method | Boston → Singapore | Dry → Rain | nuScenes→Lyft |
|---|---|---|---|
| LSS (Philion & Fidler, 2020) Baseline | 17.5 | 27.9 | 20.6 |
| Baseline + naïve LiDAR supervision | 17.5 (-0.0) | 28.2 (+0.3) | 20.3 (-0.2) |
| CroMA (Full Model) | **20.5 (+3.0)** | **29.6 (+1.7)** | **24.4 (+3.9)** |

For semantic segmentation, we report the IoU for vehicle class. To remove the effect of depth estimation, we use a pre-trained depth estimation model and fix its weight. For this task the Lidar teacher supervision refers to $L_T$ in our main pipeline. We observe that the CroMA architecture also improves the baseline method by 16.7%.

**Results of Naïve LiDAR Supervision Under Other Adaptation Settings** In Table 12, we present the ablation study of direct Lidar supervision under different adaptation scenarios. We observe that in all scenarios, naïve Lidar supervision cannot lead to better performance against the baseline.

When the source and target domains have large visual gaps (Day→Night, nuScenes→Lyft), the naive supervision leads to worse results, and when the gap is smaller (Boston→Singapore, Dry→Rain), the Lidar supervision results in on-par or only slightly better performance. The reason is that although the Lidar sensor in the source domain provides 3D knowledge to the model, it also increases the domain discrepancy between the source and the target (the model has to adapt to the additional modality shift), which hurts the model performance instead.

## D VISUALIZATION ON FAILURE CASES

As shown in Figure 6, we visualize the failure cases of our CroMA model. We can notice that most failure cases come from far distance, or heavy occlusion (which are typical failure cases for baselines as well). Faraway objects are known to be typically hard cases for monocular 3D perception (Xu & Chen, 2018; Wang et al., 2019; Brazil & Liu, 2019; Wang et al., 2021; 2022), because the ambiguity of object depth from images becomes larger as the distance increases. This can be potentially alleviated by using a smaller downsampling rate when extracting the image features. As for objects inside the red dashed boxes in Figure 6, they can be seen in the LiDAR sensor due to the higher deployment position. But in cameras, the objects are almost invisible due to the occlusion of vegetation, structures, or other vehicles. The occlusion problem can be potentially addressed if we have access to additional sensors during evaluation. As future work, we will also try to alleviate the occlusion problem in monocular settings by leveraging temporal information.

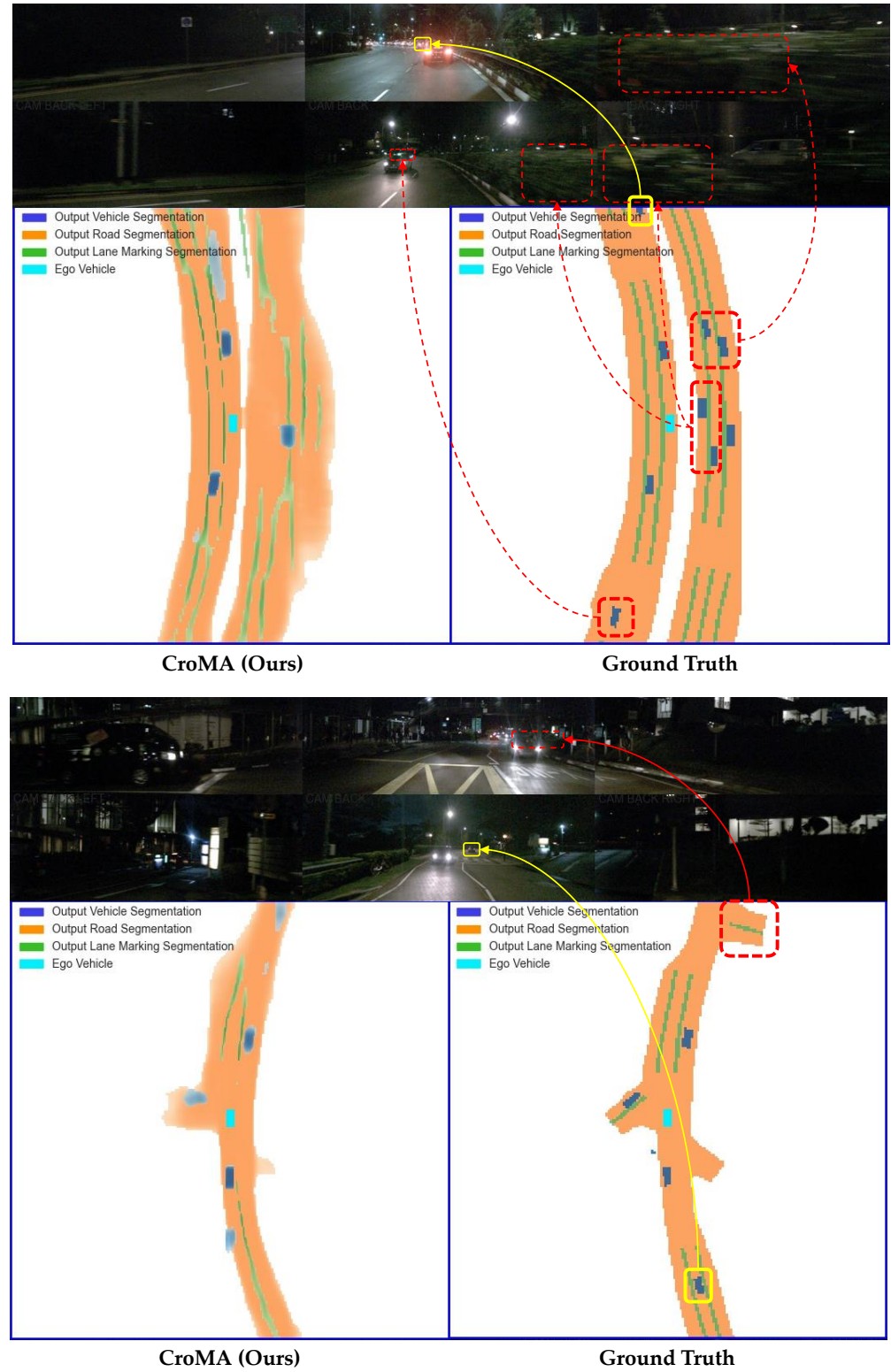

Figure 6: **Visualization of failure cases of CroMA** (model is trained with daytime data, and validated with night data). We notice that major failure cases of CroMA are Far distance and Occlusions of objects and regions.

