# OpenReview forum: "CroMA: Cross-Modality Adaptation for Monocular BEV Perception"
_ICLR.cc/2023/Conference — Submitted to ICLR 2023_

### Official Review · Reviewer_DeZJ · 2022-10-24

**Confidence:** 4
**Correctness:** 3
**Technical Novelty And Significance:** 2
**Empirical Novelty And Significance:** 2
**Recommendation:** 5

**Clarity, Quality, Novelty And Reproducibility:**

This paper clarifies its contribution well and provides details to reproduce the experiments. The contributions are somewhat new. Some of the claims need to be validated by complementary experiments.

**Strength And Weaknesses:**

Strength:
1.	Providing diagrams highlighting the differences between the proposed cross-modality adaptation and the existing fixed-modality adaptation.
2.	Performing experiments validating aspects of their claims.

Weaknesses:
1.	The proposed cross-modality adaptation has been validated in BEVDepth [1], 2DPASS [2]. This paper did not discuss them in related work.
2.	This paper proposes new dataset split setting to investigate the cross-domain perception and cross-modality adaptation, and re-runs the baselines on the proposed custom setting, making the comparisons less convinced.
3.	This paper claims that naïve LiDAR supervision leads to worse performance, which is counter-intuitive. But the paper only validates the claim under Day-to-Night Adaptation. How about City-to-City Adaptation, Dry-to-Rain Adaptation, and Dataset-to-Dataset Adaptation?
4.	This paper seems to change LSS [3] from 6 perspective camera setting to monocular setting.

[1] Li, Yinhao et al. “BEVDepth: Acquisition of Reliable Depth for Multi-view 3D Object Detection.” ArXiv abs/2206.10092 (2022): n. pag.
[2] Yan, Xu et al. “2DPASS: 2D Priors Assisted Semantic Segmentation on LiDAR Point Clouds.” ArXiv abs/2207.04397 (2022): n. pag.
[3] Philion, Jonah and Sanja Fidler. “Lift, Splat, Shoot: Encoding Images From Arbitrary Camera Rigs by Implicitly Unprojecting to 3D.” ArXiv abs/2008.05711 (2020): n. pag.


**Summary Of The Paper:**

This paper proposes a CroMA framework to incorporate multiple sensor modalities and close the domain gap between training and deployment for self-driving. They utilize LiDAR sensor during the training phase with knowledge distillation paradigm to enhance the camera-only model. Adversarial learning is adopted to address the issues of domain gap. They perform experiments to justify aspects of their claims.

**Summary Of The Review:**

This paper seems to be a step to address the issues of cross-domain perception and cross-sensor adaptation. However, there are missing complementary experiments that I feel are required to fully validate the claims. The comparisons are performed on fully customized setting, which damages the credibility of some of the claims. The cross-modality adaptation has been validated in previous works, which should be clearly discussed in the related work. Therefore, I am leaning towards a reject for this version of the work.

---

> ### Author Response · Authors · 2022-11-18
> **Response to Reviewer DeZJ (part 1)**
>
> We appreciate the reviewer for the insightful comments and suggestions. Please find our point-by-point response as follows.
>
> - **Missing related work BEVDepth and 2DPASS**:
>
>   Thanks for pointing out the related work 2DPASS [Ref1] and BEVDepth [Ref2]. We will cite them and include the discussions in the revision. 2DPASS explores how images during training can help the Lidar model to perform the 3D semantic segmentation task during inference. BEVDepth proposes to achieve reliable depth estimation for 3D object detection by exploiting the camera parameters together with the image features during depth estimation. Our work fundamentally differs from them in both the problem formulation and the methodology.
>
>   - **Difference in Problem Formulation**: Both 2DPASS and BEVDepth explore how images during training can help the Lidar model to perform the 3D segmentation or detection tasks during inference. On the contrary, we propose to use Lidar during training to help the camera model in 3D perception. We believe that our cross-modality problem – using cameras during inference as opposed to Lidar in 2DPASS and BEVDepth – is more realistic in the autonomous driving applications and also more challenging. In fact, cameras are the most *commonly equipped* sensors for autonomous vehicles, and using cameras to perceive the 3D surrounding environment is way more challenging than using Lidar due to the absence of depth-dimension information. Moreover, the cross-domain scenario and the cross-modality scenario always **appear together** in reality. Both 2DPASS and BEVDepth (as well as most existing models) address the two problems separately, while we explore their joint occurrence, which is a much harder problem.
>
>   - **Difference in Methodology**: Because of different problem settings, 2DPASS and BEVDepth are not applicable to our case. And the co-existence of domain and modality gaps in our problem further poses additional challenges to the adaptation task, as we discussed in [Sec 3.2] and Table 5 in [Sec 4.3]. In particular, naive supervision from the Lidar sensor does not help or even harms the final performance (as much as -4.5%). Our CroMA uniquely addresses this challenging setting through novel insights – as mentioned by **Reviewer Lf7W**, we “propose to leverage the depth information rather than directly using LiDAR which is intuitive and well-motivated. And LiDAR-teacher Camera-student architecture is used to bridge the gap introduced by missing modality, in both image-encoder and BEV-decoder modules.”
>
>   &nbsp;
>
>   [Ref1] Xu Yan, Jiantao Gao, Chaoda Zheng, Chao Zheng, Ruimao Zhang, Shuguang Cui, Zhen Li. 2DPASS: 2D Priors Assisted Semantic Segmentation on LiDAR Point Clouds. In ECCV, 2022.
>
>   [Ref2] Yinhao Li, Zheng Ge, Guanyi Yu, Jinrong Yang, Zengran Wang, Yukang Shi, Jianjian Sun, Zeming Li. BEVDepth: Acquisition of reliable depth for multi-view 3D object detection. In arXiv, 2022.
>
> &nbsp;
>
> - **Proposing custom adaptation settings and re-running the baselines leads to less convincing results**:
>
>   - **Direct evaluation on the no-gap scenario:** We would like to note that in Table 4, we also validate CroMA in the no-gap scenario to compare with the baselines using their reported numbers in their original paper. As we see in Table 4, CroMA also achieves state-of-the-art performance in the no-gap setting.
>
>   - **Cross-domain settings follow prior work**: We follow existing Lidar-based domain adaptation work to create the domain split strategies, including SRDAN [Ref3], ST3D [Ref4], UDA3D [Ref5], and xMUDA [Ref6]. Specifically, for Day→Night, City→City, and Dry→Rain, as described in prior approaches, we use the sentence in the nuScenes dataset and filter the keywords to split the dataset into corresponding subsets. For Dataset→Dataset split, we use the official split of the nuScenes dataset, and the split provided in ST3D [Ref4] for the Lyft dataset.
>
>   - Because we propose a new cross-modality formulation, there is no existing setting and reported numbers to compare against. However, we make sure to follow the publicly released code for baselines for a fair comparison. We will include these details in the revision and open source our code upon acceptance.
>
>   &nbsp;
>
>   [Ref3] Weichen Zhang, Wen Li, and Dong Xu. Srdan: Scale-aware and range-aware domain adaptation network for cross-dataset 3D object detection. In CVPR, 2021.
>
>   [Ref4] Jihan Yang, Shaoshuai Shi, Zhe Wang, Hongsheng Li, and Xiaojuan Qi. St3d: Self-training for unsupervised domain adaptation on 3D object detection. In CVPR, 2021.
>
>   [Ref5] Zhipeng Luo, Zhongang Cai, Changqing Zhou, Gongjie Zhang, Haiyu Zhao, Shuai Yi, Shijian Lu, Hongsheng Li, Shanghang Zhang, and Ziwei Liu. Unsupervised domain adaptive 3d detection with multi-level consistency. In ICCV, 2021.
>
>   [Ref6] Maximilian Jaritz, Tuan-Hung Vu, Raoul de Charette, Emilie Wirbel, and Patrick Perez. xmuda: Cross-modal unsupervised domain adaptation for 3D semantic segmentation. In CVPR, 2020.

---

> > ### Author Response · Authors · 2022-11-18
> > **Response to Reviewer DeZJ (part 2)**
> >
> > - **Results of naïve LiDAR supervision under other adaptation settings**:
> >
> >   Here we present the ablation study of direct Lidar supervision under different adaptation scenarios. We observe that in all scenarios, naive Lidar supervision cannot lead to better performance against the baseline.
> >
> >   When the source and target domains have large visual gaps (Day→Night, nuScenes→Lyft), the naive supervision leads to worse results, and when the gap is smaller (Boston→Singapore, Dry→Rain), the Lidar supervision results in on-par or only slightly better performance. The reason is that although the Lidar sensor in the source domain provides 3D knowledge to the model, it also increases the domain discrepancy between the source and the target (the model has to adapt to the additional modality shift), which hurts the model performance instead.
> >
> >   | Method | Boston → Singapore | Dry → Rain | nuScenes → Lyft |
> >   | :---        |    :----:   |          :----: |  :----:  |
> >   | Lift-Splat Baseline [Ref7] | 17.5 | 27.9   |  20.6 |
> >   | Baseline  + naïve LiDAR supervision | 17.5 (-0.0) | 28.2 (+0.3)      | 20.3 (-0.2) |
> >   | CroMA (Full Model) | **20.5 (+3.0)** | **29.6 (+1.7)**     | **24.4 (+3.9)** |
> >
> >   &nbsp;
> >
> >   [Ref7] Jonah Philion and Sanja Fidler. Lift, splat, shoot: Encoding images from arbitrary camera rigs by implicitly unprojecting to 3d. In ECCV, 2020.
> >
> > &nbsp;
> >
> > - **Details regarding LSS**:
> >
> >   We apologize for the confusion. We follow LSS [Ref7] to use the 6-camera setting for a fair comparison, and to perceive the whole 360 surroundings of the ego-vehicle.
> >
> >   More specifically, LSS is a camera-only model that achieves great BEV segmentation performance without domain shift. Our method differs from it in the fundamental problem setting – we propose a robust perception model that works in the co-existence of domain and modality gaps. As we described in [Sec. 4], our camera-student model follows the backbone of LSS.

---

### Official Review · Reviewer_Lf7W · 2022-10-24

**Confidence:** 4
**Correctness:** 3
**Technical Novelty And Significance:** 2
**Empirical Novelty And Significance:** 3
**Recommendation:** 5

**Clarity, Quality, Novelty And Reproducibility:**

Clarity:
- Paper is well written in general. One thing I would like to note is that though the authors mentioned that “we provide a brief theoretical insight” in the beginning of Sec.3.2, I did not feel much theory is shared.

Quality:
- This paper is of good quality as the authors provide good motivation and showcase SOTA performance. Together with the cons I described above, I have the following concerns:
1. It seems to me there are two domain gaps in this task, one in semantic segmentation space and one in depth prediction space. I did not see the ablations in these two spaces, individually. What if some generic depth estimator would give us some depth prediction? Can the results be improved when combined with existing DA methods for semantic segmentation in perspective view?
2. Why the L2 loss between BEV features extracted by student and teacher networks is a good idea? What is truly going on when training the entire model? Can the author provide some insights?
3. How are the weights in Eq.6 learnt?
4. How can the proposed method avoid degenerate cases? For instance, what if all model parameters that shared by teacher and student models are the same and the discriminator is weak?

Novelty:
- It addresses an interesting yet novel problem.

Reproducibility:
- The authors provided enough details in supplementary and code bases are available in existing work. This paper can be re-produce with some time. But other details, such as dataset split in some settings or number of cameras uses are missing.

**Strength And Weaknesses:**

Pros:
1. Well motivated and very interesting problem
2. For semantic BEV prediction, adapting depth rather than directly adapting LiDAR is intuitive and well-motivated.
3. Paper itself is well written and easy to follow

Cons:
1. Scale problem in depth prediction: Depth prediction itself suffers a lot with scale problem. I have concerns about the reliability of predicted depth during test time and am wondering why the proposed method can solve the scale problem thus bridge the gap between two domains.
2. Sparsity in LiDAR: Given the fact that LiDAR points are usually sparse, how does the proposed method handle the sparsity in depth prediction? Will the feature aggregation process be affected by the sparse depth?
3. Others and more details see below

**Summary Of The Paper:**

This paper proposes a new domain adaptation setting where multi-modalities are provided in source domain while only visual images are provided in unseen target domain. Specifically, the authors focus on the BEV prediction problem where perspective image is given as input and model would output Bev semantics. Assuming that images and LiDAR are known in source domain, the goal of this paper is to achieve accurate Bev prediction in unseen target domain that only images are provided.

To achieve that, the authors propose to leverage the depth information rather than directly using LiDAR. And LiDAR-teacher Camera student architecture is used to bridge the gap introducing by missing modality, in both image-encoder and Bev-decoder modules.

To validate their ideas, the authors conduct experiments on two datasets, NuScenes and Lyft, with four shift settings. Results on these settings show that the proposed method achieves better segmentation results compared to segmentation only methods, or method with simple depth or image level adaptation.

**Summary Of The Review:**

Overall, I feel this paper is well written and well motivated. But more explanations and experiments should be done to support the claims of the authors.

---

> ### Author Response · Authors · 2022-11-18
> **Response to Reviewer Lf7W (part 1)**
>
> We appreciate the reviewer for the insightful comments and suggestions. Please find our point-by-point response as follows.
>
> - **Why can the proposed method solve the scale problem and thus bridge the gap between two domains?**:
>
>   - We solve this problem by using the random cropping, scaling augmentation strategy during training, and also by matching the FoV (Field-of-View) of two domains using their intrinsic matrices. The augmentation increases the robustness of the depth prediction model in scale difference. And the FoV matching scales the images in the target domain to match the FoV of the source domain. This makes sure that one object looks approximately the same size in images, if it is of the same distance to the ego vehicle in source and target domains, thus reducing the scale ambiguity.
>
>     Here we provide an ablation study of scale augmentation and FoV matching in nuScenes → Lyft adaptation. As shown in the table, they both improve our final model performance.
>
>     | Model | Scale Aug. | Match FoV     |  Vehicle IoU  |
>     | :---        |    :----:   |          :----:   |   :----:   |
>     | None |         |      |    23.5  |
>     | w/ Scale Aug. | $\checkmark$ |     |    23.8 |
>     | w/ Both  |  $\checkmark$ |  $\checkmark$ |  24.4 |
>
>   - From the table, we also notice that even without the two methods mentioned above, the model still performs fairly well, compared with the baseline methods in Table 3 in the manuscript. This is because in driving scenarios, the camera FoV, the context in the images and the depth distribution of the images have a **relatively strong prior** – they do not have a strong discrepancy even across different driving scenarios (domains). This is different to more general depth estimation scenarios, where objects can have drastically different depth distribution and the intrinsic matrix can have big differences from image to image.
>
> &nbsp;
>
> - **How does the proposed method handle the sparsity of the Lidar sensor? Will the feature aggregation process be affected by the sparse depth?**
>
>   The feature aggregation process won’t be affected by the sparse nature of Lidar for these two reasons:
>
>   - The feature map is downsampled by 8 times compared with the original input image, and we crop the sky region on the top of the image. So the point density of our depth feature map is very high. We provide statistics about the sparsity: Directly projecting Lidar points onto the image plane in the original resolution results in a pixel-point coverage (ratio of pixels with at least one point projected onto them) of **less than 10%**. However, after downsampling and cropping, the pixel-point coverage ratio is **over 95%**. So the ground truth depth supervision from the Lidar is not sparse.
>
>   - There is a softmax layer to convert the depth bin estimation into a distribution, so the result will almost always be nonzero. So the estimated depth is also not sparse.
>
> &nbsp;
>
> - **Do not feel much theory is shared in Sec 3.2.**
>
>   Thanks for your valuable suggestion. We would like to clarify that in this part, we did not try to prove the error bound, but to explain the inherent trade off between the 3D information and the modality gaps brought by the Lidar sensor in our task setting, thus showing the underlying challenge of the problem. By doing so, we intended to provide another perspective to understand the challenges brought by the co-existence of cross-modality and cross-domain shifts. We will clarify it in the revision and move this part to the Appendix.

---

> > ### Author Response · Authors · 2022-11-18
> > **Response to Reviewer Lf7W (part 2)**
> >
> > - **Details including dataset split and number of cameras are missing**:
> >
> >   - We use all 6 cameras provided in the nuScenes dataset to perceive the whole 360 surroundings of the ego-vehicle.
> >
> >   - We follow existing Lidar-based domain adaptation work to create the domain split strategies, including SRDAN [Ref1], ST3D [Ref2], UDA3D [Ref3], and xMUDA [Ref4]. Specifically, for Day→Night, City→City, and Dry→Rain, as described in prior approaches, we use the sentence in the nuScene dataset and filter the keywords to split the dataset into corresponding subsets. For Dataset→Dataset split, we use the official split of the nuScenes dataset, and the split provided in ST3D [Ref2] for the Lyft dataset.
> >
> >   - We will include these details in the revision and open source our code upon acceptance.
> >
> >   &nbsp;
> >
> >   [Ref1] Weichen Zhang, Wen Li, and Dong Xu. Srdan: Scale-aware and range-aware domain adaptation network for cross-dataset 3D object detection. In CVPR, 2021.
> >
> >   [Ref2] Jihan Yang, Shaoshuai Shi, Zhe Wang, Hongsheng Li, and Xiaojuan Qi. St3d: Self-training for unsupervised domain adaptation on 3D object detection. In CVPR, 2021.
> >
> >   [Ref3] Zhipeng Luo, Zhongang Cai, Changqing Zhou, Gongjie Zhang, Haiyu Zhao, Shuai Yi, Shijian Lu, Hongsheng Li, Shanghang Zhang, and Ziwei Liu. Unsupervised domain adaptive 3d detection with multi-level consistency. In ICCV, 2021.
> >
> >   [Ref4] Maximilian Jaritz, Tuan-Hung Vu, Raoul de Charette, Emilie Wirbel, and Patrick Perez. xmuda: Cross-modal unsupervised domain adaptation for 3D semantic segmentation. In CVPR, 2020.
> >
> > &nbsp;
> >
> > - **The ablation in the semantic segmentation space domain gap and the depth prediction space domain gap**:
> >
> >   This is an interesting perspective of the domain shift. Here we show the result of depth estimation and semantic segmentation alone in the Day → Night scenario.
> >
> >   - For depth estimation, we report the cross entropy error (CEE) with the ground truth we described in Sec 3.2, because each pixel will have multiple depth values. For this task the Lidar teacher supervision refers to $L_{dp}$ in our main pipeline. We observe that the method we propose significantly improves the depth estimation metrics by 44.9%.
> >
> >   - For semantic segmentation, we report the IoU for vehicle class. To remove the effect of depth estimation, we use a pre-trained depth estimation model and fix its weight. For this task the Lidar teacher supervision refers to $L_{T}$ in our main pipeline. We observe that the CroMA architecture also improves the baseline method by 16.7%.
> >
> >     | Model | Depth Estimation (CEE) | Semantic Segmentation (IoU)     |
> >     | :---        |    :----:   |          :----: |
> >     | Direct Inference | 2.56 | 27.5   |
> >     | Adversarial Learning (AL) | 1.97  | 31.8  |
> >     | Lidar-Teacher Supervision + AL | 1.41  | 32.1  |
> >
> >   However, depth estimation and semantic segmentation are heavily **correlated** in our task, where BEV semantic segmentation is the final objective, and depth estimation is the intermediate representation that bridges the 2D view with the BEV view. Thus, learning a better depth estimation naturally leads to better overall segmentation, and learning a better segmentation task also leads to better depth estimation by the backpropagate of gradient, as demonstrated in Table 5.
> >
> >   If a great generic depth estimator provides us with better cross-domain depth estimation, it will improve the final performance. But it won’t eliminate the importance of the Lidar teacher final-layer supervision, as we discussed in Sec. 4.2. We also believe it is possible to transfer the knowledge from existing perspective-view semantic segmentation DA models. However, it is beyond the scope of this work, and we leave it as interesting future directions.
> >
> > &nbsp;
> >
> > - **Why is the loss on the feature map a good idea? Some insights are needed.**
> >
> >   - The goal of the teacher model is to transfer the knowledge of Lidar into the student model, without asking the student to directly take Lidar as input. The final-layer feature map is a good learning objective, because in the teacher model it encodes the camera and Lidar sensor information, and its gradients can be back-propagated to supervise the whole student model.
> >
> >   - In addition to the feature map supervision, another common supervision comes from teacher labels. It uses the teacher model’s final class output (a distribution vector) to supervise the student model. We call this “Soft-label Supervision.” As shown in Appendix Table 6, feature-level supervision performs ~15% better than soft-label supervision. We also provide some analysis. One main reason is that compared with commonly used 91-class COCO and 200-class ImageNet training, we only train binary class segmentation. And with few classes, the teacher’s final label distribution does not hold much semantic information, hence is less helpful, while the feature map still preserves a rich portion of meaningful information.

---

> > > ### Author Response · Authors · 2022-11-18
> > > **Response to Reviewer Lf7W (part 3)**
> > >
> > > - **How are the weights in Eq.6 learnt?**
> > >
> > >   The weights are hyperparameters that are not learned by the model. We use the weight factors to control the loss terms only such that they have approximately the same order of magnitude. The weights are set using the validation set and the values are provided in the Appendix.
> > >
> > > &nbsp;
> > >
> > > - **How can the proposed method avoid degenerate cases like identical teacher and student, and the weak discriminator?**
> > >
> > >   - We optimize a joint loss as our learning objective, and the loss weight $\lambda_T$ controls the supervision strength of the teacher model and we use this to avoid the model degeneration. As we see in the table below, when the weight of the teacher supervision increases, it hurts the final performance because the student model becomes more identical to the teacher model without considering the proper domain shift between the source and target.
> > >
> > >     |  | Vehicle | Road | Lane |
> > >     | :---        |    :----:   |          :---: | :---: |
> > >     | $\lambda_T=0.1$      | 16.7 | 50.6   | 16.2    |
> > >     | $\lambda_T=1.0$   | **17.0** | **51.8**      |  **16.9** |
> > >     | $\lambda_T=10.0$   | 11.1 | 38.8      |  10.5 |
> > >     | $\lambda_T=100.0$   | 8.3 | 26.0      |  6.8 |
> > >
> > >   - As for the discriminator, we simply use a global average pooling layer followed by two fully connected layers to generate the domain label. As we see in the table below, our model is not sensitive to the weight of the discriminator loss, and we show that such a light-weight discriminator is easy to optimize for and is enough to distinguish different domains and guide the learning of domain-agnostic features.
> > >
> > >     | | Vehicle | Road | Lane |
> > >     | :---        |    :----:   |          :---: | :---: |
> > >     | $\lambda_{D_1}=0.02$      | 16.9 | 51.3   | 16.8    |
> > >     | $\lambda_{D_1}=0.1$   | **17.0** | **51.8**      |  **16.9** |
> > >     | $\lambda_{D_1}=0.5$   | 16.8 | 51.8      |  16.4 |
> > >     | $\lambda_{D_1}=1.0$   | 15.9 | 49.5      |  16.0 |

---

### Official Review · Reviewer_Dv6y · 2022-10-24

**Confidence:** 4
**Clarity, Quality, Novelty And Reproducibility:** This paper has good clarity, moderate…
**Correctness:** 3
**Technical Novelty And Significance:** 2
**Empirical Novelty And Significance:** 2
**Recommendation:** 5

**Strength And Weaknesses:**

---
Strengths:
* The paper is well-written and easy to follow. The authors have provided sufficient background on monocular BEV perception, which is helpful for readers outside this domain.
* The proposed solution is technically sound and achieves good empirical performance on domain adaptation.

---
Weaknesses:
* The technical novelty is very limited. The proposed techniques in this paper are not new: (1) 2DPASS has explored cross-modality knowledge distillation, and (2) many domain adaptation methods apply adversarial learning. Combining existing methods is a good engineering effort but cannot be considered a solid contribution.
* All experimental results target BEV segmentation. It is essential to present some results on 3D object detection benchmarks.
* Despite the good overall paper writing, I do not quite appreciate the theoretical insights provided in Section 3.2. The domain error bound seems far-fetched and not very related to the proposed method.
* Section 3.1 describes the exact method from LSS without any proper citation.

---

**Summary Of The Paper:**

This paper aims to transfer the point cloud knowledge from a LiDAR sensor during the training phase to the camera-only testing scenario. The authors apply knowledge distillation with a LiDAR teacher and a camera student. They also use the multi-level adversarial learning mechanism to adapt the features learned from different sensors and domains. The proposed CroMA delivers fairly good domain adaptation performance.

**Summary Of The Review:**

The current recommendation is primarily based on the limited novelty and insufficient evaluation. However, I would love to see the authors' response before making the final recommendation.

---

> ### Author Response · Authors · 2022-11-18
> **Response to Reviewer Dv6y (part 1)**
>
> We appreciate the reviewer for the insightful comments and suggestions. Please find our point-by-point response as follows.
>
> - **Related work 2DPASS, and concern about novelty**:
>
>   We respectfully disagree with the reviewer that 1) our Lidar-Teacher and Camera-Student knowledge distillation is similar to the cross-modality knowledge distillation in 2DPASS [Ref1]; and 2) our work is just a good engineering effort. We clarify our novelty as follows:
>
>   - First, thanks for pointing out the related work 2DPASS [Ref1], and we will cite it and include the discussion in the revision. Though 2DPASS also explores the cross-modality model learning, our work fundamentally differs from it in both the problem formulation and the methodology.
>
>     - **Our Novelty in Problem Formulation**: 2DPASS explores how images during training can help the Lidar model to perform the 3D semantic segmentation task during inference. On the contrary, we propose to use Lidar during training to help the camera model in 3D perception. We believe that our cross-modality problem – using cameras during inference as opposed to Lidar in 2DPASS – is more realistic in the autonomous driving applications and also more challenging. In fact, cameras are the most *commonly equipped* sensors for autonomous vehicles, and using cameras to perceive the 3D surrounding environment is way more challenging than using Lidar due to the absence of depth-dimension information. Moreover, the cross-domain scenario and the cross-modality scenario always *appear together* in reality. 2DPASS and most existing models address the two problems separately, while we explore their joint occurrence, which is a much harder problem.
>
>     - **Our Novelty in Methodology**: Because of different problem settings, 2DPASS is not applicable to our case. And the co-existence of domain and modality gaps in our problem further poses additional challenges to the adaptation task, as we discussed in Sec. 3.2, Table 5 in Sec. 4.3 and the reply to reviewer DeZJ. In particular, naive supervision from the Lidar sensor does not help or even harms the final performance (as much as -4.5%). Our CroMA uniquely addresses this challenging setting through novel insights – as mentioned by **Reviewer Lf7W**, we “propose to leverage the depth information rather than directly using LiDAR which is intuitive and well-motivated. And LiDAR-teacher Camera-student architecture is used to bridge the gap introduced by missing modality, in both image-encoder and BEV-decoder modules.”
>
>   - We would like to further argue that exploiting existing techniques does not hinder the novelty of our work. As mentioned above, to the best of our knowledge, CroMA is the first work that aims to loosen the constraints in the BEV scene segmentation task by allowing the co-existence of the domain gap and the sensor gap between training and inference phases. As mentioned by **reviewer 5QJE**, CroMA “*contributes to the robust estimation of 3D scene representation in BEV under both domain shift and modality change*,” which is a vital topic yet missing in our community right now. Conventional adversarial methods cannot exploit the source-only Lidar, and intriguingly, our work shows that the naive Lidar supervision leads to worse final performance (Table 5, -4.5%). On the contrary, we explore and analyze the **synergy** of knowledge distillation and the multi-level adversarial learning in solving four different adaptation scenarios, and achieve up to 14.1% improvement over the best baselines – such synergy provides a novel perspective to address the co-existence of domain and sensor gaps. We also propose a new progressive adaptation strategy to address the challenging mixed domain gap scenario in Sec. 4.3, and achieve up to 71.4% improvement compared with the direct adversarial learning method.
>
>   &nbsp;
>
>   [Ref1] Xu Yan, Jiantao Gao, Chaoda Zheng, Chao Zheng, Ruimao Zhang, Shuguang Cui, Zhen Li. 2DPASS: 2D Priors Assisted Semantic Segmentation on LiDAR Point Clouds. In ECCV, 2022.

---

> > ### Author Response · Authors · 2022-11-18
> > **Response to Reviewer Dv6y (part 2)**
> >
> > - **Results in 3D object detection**:
> >
> >   Thanks for the suggestion. In principle, our work is general and applicable to different types of BEV perception tasks.
> >
> >   - We chose to perform BEV scene segmentation following existing work MonoLayout [Ref2], OFT [Ref3], LSS [Ref4], and because this task provides a more thorough understanding of the surrounding environment of the ego-agent not limited to vehicles, but also including estimation of roads and lane dividers.
> >
> >   - As mentioned by **reviewer 5QJE**, We did “extensive experiments” for the BEV segmentation task *in 4 different domain and modality shift settings, no-shift setting and mixed-shift setting*. We believe that our current evaluation validates the effectiveness of our approach.
> >
> >   - We agree with the reviewer that the evaluation would be more comprehensive if we include results on 3D object detection. We are currently experimenting with this task, and modifying the codebase for detection head and re-training the model for the new task require some additional time. We will include the results in the final manuscript. If time allows, we will provide them by the end of the discussion period.
> >
> >   &nbsp;
> >
> >   [Ref2] Kaustubh Mani, Swapnil Daga, Shubhika Garg, Sai Shankar Narasimhan, Madhava Krishna, and Krishna Murthy Jatavallabhula. Monolayout: Amodal scene layout from a single image. In WACV, 2020.
> >
> >   [Ref3] Thomas Roddick, Alex Kendall, and Roberto Cipolla. Orthographic feature transform for monocular 3D object detection. In BMVC, 2019
> >
> >   [Ref4] Jonah Philion and Sanja Fidler. Lift, splat, shoot: Encoding images from arbitrary camera rigs by implicitly unprojecting to 3D. In ECCV, 2020.
> >
> > &nbsp;
> >
> > - **“theoretical insight” might be unnecessary**:
> >
> >   Thanks for your valuable suggestion. We would like to clarify that in this part, we did not try to prove the error bound, but to explain the inherent trade off between the 3D information and the modality gaps brought by the Lidar sensor in our task setting, thus showing the underlying challenge of the problem. By doing so, we intended to provide another perspective to understand the challenges brought by the co-existence of cross-modality and cross-domain shifts. We will clarify it in the revision and move this part to the Appendix.
> >
> > &nbsp;
> >
> > - **Missing citation of LSS in Section 3.1**:
> >
> >   Thanks for the reminder and correction. We cited and discussed LSS (Lift-Splat) [Ref4] in Related Work and Experiment sections of the original manuscript. We will add the citation of LSS in Section 3.1 in the revised version.

---

> > > ### Comment · Reviewer_Dv6y · 2022-11-25
> > > **Reviewer Response**
> > >
> > > Thanks for the detailed reply! I'm afraid that I have to disagree with your argument on technical novelty:
> > > * The problem setting where LiDAR is only used in the training time is not novel. In fact, it's extensively explored in previous papers, such as DD3D, PGD and BEVDepth. Based on the results in these papers, it's not really surprising that LiDAR provides complementary supervision signal to the monocular camera models.
> > > * The claim that naive depth supervision does not work seems to be contradictory to the results in these previous papers. I haven't found an intuitive way to understand this discrepancy.
> > >
> > > Therefore, I will keep my original score based on the current discussion.

---

> > > > ### Author Response · Authors · 2022-11-27
> > > > **Response to Reviewer Dv6y (Follow-up questions)**
> > > >
> > > > Thanks for your response. Regarding your concerns:
> > > >
> > > > ---
> > > > > **Q1:** Using LiDAR during training time is not novel and is explored in DD3D, PGD and BEVDepth.
> > > >
> > > > We do not claim that our method is the first in exploiting LiDAR during training time. On the contrary, we emphasize that the common practice of exploiting the *“complementary”* Lidar during training leads to worse performance **when training and testing data reside in different domains**.
> > > >
> > > > DD3D [Ref1], PGD [Ref2], and BEVDepth [Ref3] use Lidar as supervision for the image depth estimation and prove that Lidar introduces useful knowledge to help 2D→3D understanding. What these prior methods are missing is a more realistic setting where training and testing happen in different domains. In this case, the knowledge learned from the Lidar in the source domain does not naturally transfer to the target domain, which leads to inconsistent behavior of target domain inference. Compared with prior work, we are unique in exploiting the right technique to retain the Lidar supervision knowledge between domain shifts. We will add the citation of the related work and include the discussion in the final manuscript.
> > > >
> > > > ---
> > > >
> > > > > **Q2:** The claim that naive depth supervision does not work seems to be contradictory to the results in these previous papers. I haven't found an intuitive way to understand this discrepancy
> > > >
> > > > As we explained in **Q1**, the contradiction with the previous papers regarding the usefulness of Lidar supervision comes from the domain shift. Lidar introduces useful knowledge to help depth estimation and thus 2D→3D understanding, but the knowledge from the source domain does not naturally transfer to the new domain. For example, the knowledge for better estimating depth in the daylight may not help the depth estimation for night scenes. According to Table 12 in the appendix (which we also list below), the influence of the source domain Lidar supervision is inconsistent among different domain shifts. But with our proposed Lidar-Teacher supervision and the multi-level adversarial learning, the knowledge from the Lidar supervision is able to provide consistent improvement to the target domain monocular model.
> > > >
> > > >   | Method | Boston → Singapore | Dry → Rain | nuScenes → Lyft |
> > > >   | :---        |    :----:   |          :----: |  :----:  |
> > > >   | Lift-Splat Baseline [Ref7] | 17.5 | 27.9   |  20.6 |
> > > >   | Baseline  + naïve LiDAR supervision | 17.5 (-0.0) | 28.2 (+0.3)      | 20.3 (-0.2) |
> > > >   | CroMA (Full Model) | **20.5 (+3.0)** | **29.6 (+1.7)**     | **24.4 (+3.9)** |
> > > >
> > > >
> > > > ---
> > > >
> > > > #### **Reference**
> > > >
> > > > [Ref1] Dennis Park, Rares Ambrus, Vitor Guizilini, Jie Li, Adrien Gaidon. Is Pseudo-Lidar needed for Monocular 3D Object detection? In ICCV, 2021.
> > > >
> > > > [Ref2] Tai Wang, Xinge Zhu, Jiangmiao Pang, Dahua Lin. Probabilistic and Geometric Depth: Detecting Objects in Perspective. In CoRL, 2021.
> > > >
> > > > [Ref3] Yinhao Li, Zheng Ge, Guanyi Yu, Jinrong Yang, Zengran Wang, Yukang Shi, Jianjian Sun, Zeming Li. BEVDepth: Acquisition of reliable depth for multi-view 3D object detection. In arXiv, 2022.

---

> ### Author Response · Authors · 2022-12-08
> **Update: Results for 3D Object Detection**
>
> We thank the reviewer for suggesting the 3D object detection task in addition to the BEV segmentation benchmark. As we explained in the previous response, we chose to perform BEV scene segmentation because this task provides a more thorough understanding of the surrounding environment of the ego-agent not limited to vehicles, but also including estimation of roads and lane dividers.
>
> Here, to answer the reviewer’s question and further test the generalizability of our proposed method, we investigated the 3D object detection benchmark. We keep the model backbone the same and use the CenterPoint [Ref1] detection head to replace our original segmentation head. We report the results in Mean-Average-Precision (mAP), nuScenes-Detection-Score (NDS), and other metrics used by the nuScenes benchmark. The comparison of the results under the Singapore → Boston adaptation setting is shown below:
>
> | Singapore → Boston      |   DA   | CM     |   mAP$\uparrow$ | NDS$\uparrow$  | mATE$\downarrow$  | mAOE$\downarrow$ |
> | :---        |    :----:   |    :----:   | :----:   | :----:   | :----:   | :----:   |
> | LSS [Ref2]      |   |      |  0.160 | 0.203 | 0.867 | 0.730 |
> | Vanilla DA  | $\checkmark$ |        |   0.184| 0.224 | 0.838 | 0.671 |
> | Depth-Supv DA  | $\checkmark$ |    $\checkmark$     |   0.191| 0.235 | 0.815 | 0.647 |
> | CroMA (ours)  | $\checkmark$ |   $\checkmark$      |   **0.225** | **0.261** | **0.798** | **0.606** |
>
> As shown in the table, although direct domain adaptation and Lidar depth supervision both boost the detection baseline, our proposed Lidar-Teacher and multi-level adversarial learning help CroMA achieve the best performance on all metrics. **The improvement on mAP over the baseline LSS is over 40% for mAP and over 28% for NDS**. This experiment again demonstrates the generalization of our proposed method across various downstream tasks. We will include this result in the final manuscript.
>
> We hope this experiment, in addition to our previous responses (posted on November 18 and 27) to the official review, can address the concerns raised by the reviewer. Please let us know if there are any other remaining questions. We would be very happy to address them.
>
> ---
>
> ### **Reference**
>
> [Ref1] Tianwei Yin, Xingyi Zhou, Philipp Krähenbühl. Center-based 3D Object Detection and Tracking. In CVPR, 2021.
>
> [Ref2] Jonah Philion and Sanja Fidler. Lift, splat, shoot: Encoding images from arbitrary camera rigs by implicitly unprojecting to 3d. In ECCV, 2020.

---

### Official Review · Reviewer_5QJE · 2022-10-31

**Confidence:** 3
**Correctness:** 4
**Technical Novelty And Significance:** 3
**Empirical Novelty And Significance:** 3
**Recommendation:** 8

**Clarity, Quality, Novelty And Reproducibility:**

The paper is clear and the description of the technical work is easy to follow and understand. The experimental validation is extensive and there is novelty in that the testing data is not annotated and there is evaluation of dataset-to-dataset adaptation. There are also results evaluation cross-domain adaptation. There is some lack of detail that may hamper the reproducibility of the results.

**Details Of Ethics Concerns:**

There are no ethics concerns.

**Strength And Weaknesses:**

The strengths of the work results from the approach that is able to learn 3D BEV representations of scenes with both domain shift and modality mismatch. Another strength is the alignment of feature space between domains. The main weakness results from the lack of results concerning the computational complexity.

**Summary Of The Paper:**

This paper proposes an approach for learning a monocular bird's-eye-view that transfers knowledge from Lidar data (used in training) to the testing scenario where only camera images are used. The application is self-driving. The approach is based on a Lidar-Teacher and Camera-Student knowledge distillation model. Therefore multiple sensor modalities are used (LiDAR and cameras) and domain adaptation is also handled. The point clouds knowledge from Lidar sensor is used only during the training phase. A multi-level adversarial learning mechanism is used to adapt and align features learned from different sensors and domains. Therefore there is cross-domain perception and cross-sensor adaptation for monocular 3D tasks.

**Summary Of The Review:**

Interesting work that handles cross-domain and cross-modality adaptation. This paper contributes to the robust estimation of 3D scene representation in BEV under both domain shift and modality change. The results show improvements in the estimation of BEV compared with competing approaches.

---

> ### Author Response · Authors · 2022-11-18
> **Response to Reviewer 5QJE**
>
> Thank you for your appreciation of our work! Please find our response to the questions about computational complexity and reproducibility as follows.
>
> - **The analysis of computational complexity**:
>
>   We provide a table that summarizes the number of parameters and inference speed for prior baselines and our model as below.
>
>   |   Model     | #Params (M) | Frame-per-Second (FPS)  |
>   | :---        |    :----:   |          :----: |
>   | OFT [Ref1]        |   22   |          25 |
>   | Lift-Splat-Shoot (LSS) [Ref2] | 14       | 35   |
>   | CroMA (Ours) | 15        | 33      |
>
>   From the table, we can see that our Lidar-Teacher distillation and multi-level adversarial learning modules do not affect the inference efficiency of CroMA. Our total number of parameters is 15M, and our inference time is 33 Frame-per-Second (FPS) on a V100 GPU, which is on par with the baseline LSS [Ref2]. The training time for our model is around 20 hours on 4xV100 GPUs. We will include this analysis of computational complexity in the revised manuscript.
>
>   &nbsp;
>
>   [Ref1] Thomas Roddick, Alex Kendall, and Roberto Cipolla. Orthographic feature transform for monocular 3D object detection. In BMVC, 2019
>
>   [Ref2] Jonah Philion and Sanja Fidler. Lift, splat, shoot: Encoding images from arbitrary camera rigs by implicitly unprojecting to 3d. In ECCV, 2020.
>
> &nbsp;
>
> - **Some lack of details regarding the reproducibility**:
>
>   We provided additional details about our model architecture and dataset split in the Appendix. We also elaborate the way we create the data split and the usage of 6 cameras in the reply to reviewers Lf7W and DeZJ. We will open-source our codebase upon acceptance.

---

### Author Response · Authors · 2022-11-19
**General Response & Comments Incorporated in the Newest Revision**

We would like to thank all the reviewers for their insightful comments and constructive suggestions. We have provided our detailed response to each reviewer. We look forward to further feedback and discussion.

We’ve also uploaded a new version of the paper that incorporates reviewers’ comments, with major changes highlighted in blue (there are also other minor changes on wording and typos). Major changes include:

- **Analysis on Computational Complexity** [Reviewer **5QJE** ]: We include a subsection on computational complexity analysis in the appendix.

- **Citation and Related Work** [Reviewers **Dv6y, DeZJ**]:  We add citation and discussion about 2DPASS and BEVDepth. We also add the missing citation of Lift-Splat (LSS) in Section 3.1.

- **Details on dataset split and training** [Reviewers **5QJE, Lf7W, DeZJ**]: We introduce in detail the dataset split setting in the appendix. We also elaborate that we are using all six cameras provided by the dataset for a fair comparison with the baselines and for a thorough perception of the 360 surroundings.

- **Unclear insight paragraph in Section 3.2** [Reviewers **Dv6y, Lf7W**]: We rephrase the paragraph to emphasize the challenge brought by the new problem setting. And we move the original paragraph into the appendix.

- **Missing ablation experiments** [Reviewers **Lf7W, DeZJ**]: We add several complementary experiments to validate the effectiveness of our model. Specifically, we include the analysis for scale ambiguity, separate domain gaps (segmentation and depth estimation), and results of naive LiDAR supervision under other adaptation settings.

---

### Author Response · Authors · 2022-12-08
**Update: General response**

We kindly remind the reviewers that the discussion stage-2 (among authors, reviewers, and AC) is about to end shortly. We hope our explanations and additional experimental results have addressed the reviewers’ concerns. If you have any further concerns or questions, please do not hesitate to let us know.

In addition to the initial replies to the official reviews (summarized in the General Response posted on November 18), here we also summarize further explanations and experimental results we conducted during the discussion phase:

- **Results on 3D object detection task** [Reviewer **Dv6y**]: We include additional results on the 3D object detection benchmark. CroMA achieves the best results on all metrics including mAP and NDS, demonstrating the generalization of our proposed method on various downstream tasks.

| Singapore → Boston      |   DA   | CM     |   mAP$\uparrow$ | NDS$\uparrow$  | mATE$\downarrow$  | mAOE$\downarrow$ |
| :---        |    :----:   |    :----:   | :----:   | :----:   | :----:   | :----:   |
| LSS Baseline      |   |      |  0.160 | 0.203 | 0.867 | 0.730 |
| Vanilla DA  | $\checkmark$ |        |   0.184| 0.224 | 0.838 | 0.671 |
| Depth-Supv DA  | $\checkmark$ |    $\checkmark$     |   0.191| 0.235 | 0.815 | 0.647 |
| **CroMA (ours)**  | $\checkmark$ |   $\checkmark$      |   **0.225** | **0.261** | **0.798** | **0.606** |

- Further explanation of the **unintuitive observation** regarding the naive depth supervision, and the **uniqueness of our problem setting**. [Reviewer **Dv6y**]

- Further explanation of the **domain gap** and **LiDAR sparsity** [Reviewers **Lf7W**]

&nbsp;

Thank you very much!

Authors

---

### Decision · Program_Chairs · 2023-01-20

**Decision:**

Reject

**Justification For Why Not Higher Score:**

Three out of the four reviewers are negative, and the fourth reviewer (while giving the work a high rating) also agrees that the paper has weaknesses and does not object to reject.  The AC feel that the paper is missing discussion and comparison of key work (MonoDistill).

**Justification For Why Not Lower Score:**

N/A

**Metareview: Summary, Strengths And Weaknesses:**

Summary: The paper proposes a Cross-Modality Adaptation (CroMA) framework for monocular BEV perception.  The work explores the setting of when there are different sensors available during training and test.  Specifically, they consider the case when LiDAR is available during training but during testing, only a RGB camera is available.  The work proposes training a Camera-student model to project BEV features with a LiDAR-teacher.  The LiDAR-teacher had depth information that allows it to directly transform images features to the BEV frame, and serves as an guide for the Camera-student.  The model is trained using an adversarial schemes to align the features of the BEV and image domains and allows for robustness to domain shifts.  Domain-shift (city-to-city, day-to-night, dry-to-rain, dataset-to-dataset) experiments are conducted on the nuSenes and Lyft datasets,  Results show that the proposed CroMA method outperforms prior methods for most of the scenarios.

Strengths
- The paper clearly explains the cross-modality setting where the sensors available during test time may be a subset of what is available during training time.  This is an important setting that should be encouraged.
- Experiments show that the proposed method can handle domain shifts
- Proposed solution is technical sound

Weaknesses
- Missing discussion and comparison against recent relevant work on distillation for 3D detection.  Upon the request of reviewers Dv6y and DeZj, the authors have added citations and discussion for 2DPASS, BEVDepth, and Lift-Splat.  However, the AC believe that there are relevant recent work using 3D-to-2D distillation that are not mentioned including:
  - MonoDistill: Learning Spatial Features for Monocular 3D Object Detection, Chong et al. ICLR 2022
  - LIGA-Stereo: Learning LiDAR Geometry Aware Representations for Stereo-based 3D Detector, Guo et al. ICCV 2021
  - Cross-Modality Knowledge Distillation Network for Monocular 3D Object Detection, Hong et al. ECCV 2022

Some of these recent methods, such as MonoDistill (which only uses the student RGB model during test time) should be included for comparison as well.
- Limited technical novelty as there has been prior work in cross-modality adaptation that the paper did not discuss in detail


**Summary Of Ac-Reviewer Meeting:**

The AC had a virtual meeting with reviewer 5QJE and Dv6y.  Reviewer 5QJE was the most positive (with a rating of 8) but also acknowledges that the paper had weakneses, especially missing discussion of recent work on 3D-to-2D distillation.  He was okay with the paper being reject as the other reviewers felt the paper was below the acceptance bar.  Reviewer Dv6y indicated concern with the limited technical novelty of the work, as prior works have explored cross-modality knowledge

Reviewer Lf7W was not able to make the meeting and stated on OpenReview that their opinion was unchanged (negative).

Reviewer DeZJ did not respond to request to offer availability nor engage in discussion on OpenReview.
Three out of the four reviewers are negative, and the fourth reviewer (while giving the work a high rating) also agrees that the paper has weaknesses and does not object to reject.  The AC feel that the paper is missing discussion and comparison of key work (MonoDistill).